# Design of complicated all-α protein structures

Koya Sakuma [1,10], Naohiro Kobayashi[2,3,10], Toshihiko Sugiki [3],
Toshio Nagashima[2], Toshimichi Fujiwara [3], Kano Suzuki [4],
Naoya Kobayashi [5], Takeshi Murata [4,6,7], Takahiro Kosugi [1,5,8],
Rie Tatsumi-Koga [5] & Nobuyasu Koga [1,5,8,9] ✉

A wide range of de novo protein structure designs have been achieved, but the complexity of naturally occurring protein structures is still far beyond these designs. Here, to expand the diversity and complexity of de novo designed protein structures, we sought to develop a method for designing 'difficult-to-describe' α-helical protein structures composed of irregularly aligned α-helices like globins. Backbone structure libraries consisting of a myriad of α-helical structures with five or six helices were generated by combining 18 helix–loop–helix motifs and canonical α-helices, and five distinct topologies were selected for de novo design. The designs were found to be monomeric with high thermal stability in solution and fold into the target topologies with atomic accuracy. This study demonstrated that complicated α-helical proteins are created using typical building blocks. The method we developed will enable us to explore the universe of protein structures for designing novel functional proteins.

Many naturally occurring protein structures are complicated, lacking distinguishable symmetry and regularity. Prominent examples of such complicated proteins are globin-fold structures with eight irregularly packed α-helices; Kendrew referred to the tertiary arrangement of the secondary structures as being difficult to describe in simple terms[1] (Fig. 1a). In most parts of globin fold structures, two helices adjacent in the sequence are connected crosswise rather than hairpin-like, and the helix–helix packings deviate from the canonical patterns[2,3]; this fold does not include internal structural repeats such as α-solenoids[4,5]. These asymmetric, irregular and nonrepetitive secondary structure arrangements make it difficult to simply describe globin structures, and many naturally occurring proteins as well.

A wide range of all-α protein structures have been designed, but the designs have been limited to simple and ordered structures consisting of α-helices in almost parallel alignment, such as coiled-coil, bundle and barrel structures (Fig. 1b–d and Extended Data Fig. 1)[5–27]. Jacobs et al. attempted to design α-helical proteins with more variety[15], but their designs were still bundle-like (the two designs with five α-helices in Fig. 1b). However, the distribution of complexity for naturally occurring all-α protein structures is biased to the complicated ones (Fig. 1d). The observed distribution bias is probably due to the fact that all-α proteins with complicated spatial arrangements of α-helices can provide diverse and heterogenous molecular surfaces, enabling specific interactions with binding partners. Moreover, such complicated all-α proteins should make it possible to incorporate a functional site enclosed on

[1]Department of Structural Molecular Science, School of Physical Sciences, SOKENDAI (The Graduate University for Advanced Studies), Hayama, Japan. [2]RIKEN Center for Biosystems Dynamics Research, RIKEN, Yokohama, Japan. [3]Institute for Protein Research, Osaka University, Suita, Japan. [4]Department of Chemistry, Graduate School of Science, Chiba University, Chiba, Japan. [5]Protein Design Group, Exploratory Research Center on Life and Living Systems (ExCELLS), National Institutes of National Sciences, Okazaki, Japan. [6]Membrane Protein Research Center, Chiba University, Chiba, Japan. [7]Structural Biology Research Center, Institute of Materials Structure Science, High Energy Accelerator Research Organization (KEK), Tsukuba, Japan. [8]Research Center of Integrative Molecular Systems, Institute for Molecular Science, National Institutes of National Sciences, Okazaki, Japan. [9]Present address: Institute for Protein Research, Osaka University, Suita, Japan. [10]These authors contributed equally: Koya Sakuma, Naohiro Kobayashi. ✉e-mail: nkoga@protein.osaka-u.ac.jp

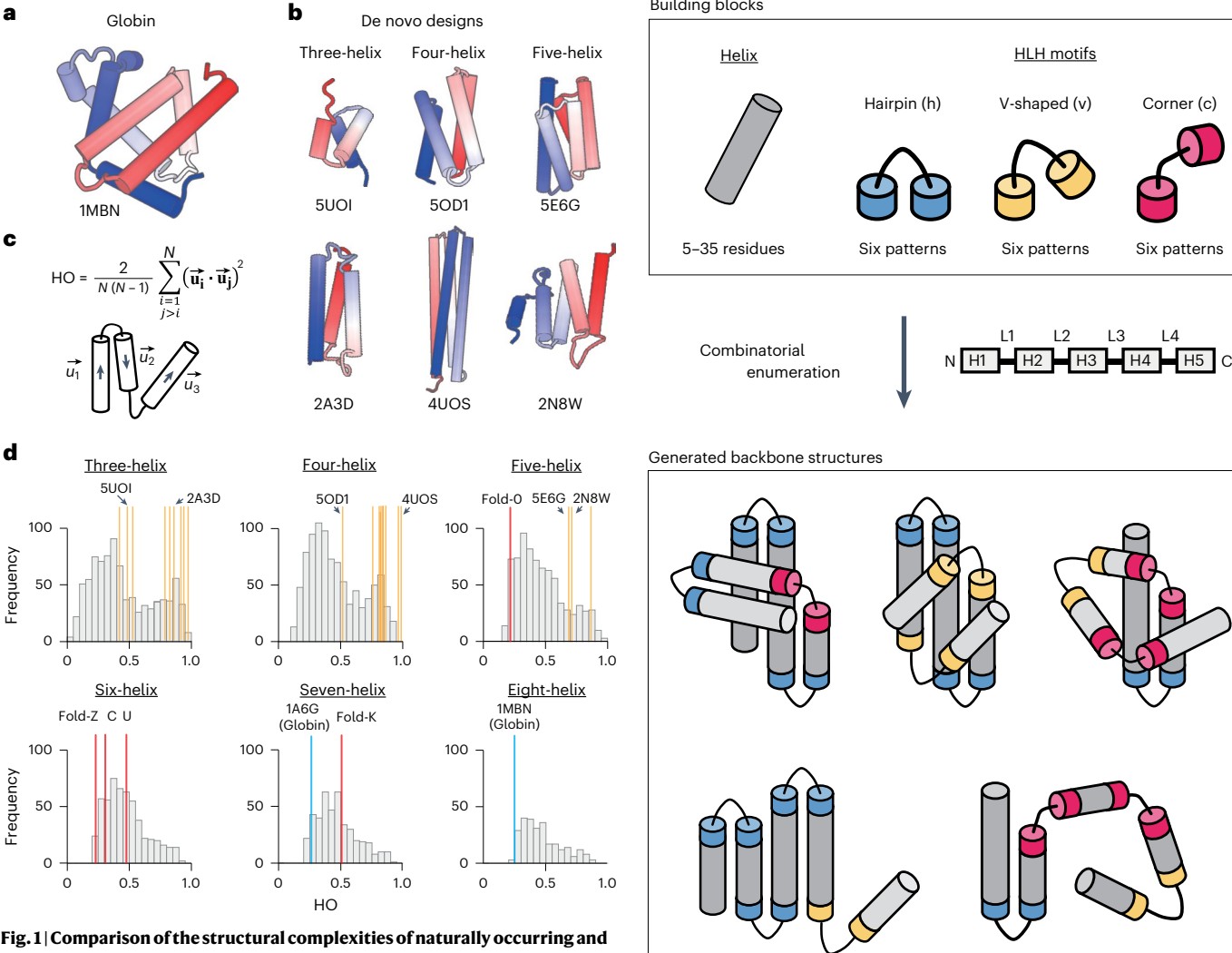

**Fig. 1 | Comparison of the structural complexities of naturally occurring and de novo designed proteins. a,b,** Structures of myoglobin (**a**) and representative de novo designed all-α proteins (**b**) (the N- and C-terminals are colored in blue and red, respectively, and the characters represent PDB IDs). The α-helices in the globin structure are irregularly aligned, whereas those of the de novo designs are almost parallelly aligned. **c,** The order parameter capturing the complexities of α-helical proteins, HO. HO is defined by the average of inner products between helix orientation vectors, $\mathbf{u}_i$, for all pairs of $N$ α-helices[55]. Higher values indicate more ordered, and lower values more complicated. **d,** HO distributions for naturally occurring and de novo designed proteins with three to eight α-helices. Whereas naturally occurring all-α proteins show broad distributions irrespective of the number of constituent α-helices, previous de novo designed all-α proteins indicated by yellow-ocher bars show relatively higher values in the distributions (for details of the previous designs, see Extended Data Fig. 1). Notably, globin structures indicated by blue bars have quite low values. The all-α proteins created in this study, indicated by red bars, have lower values than the previous designs.

**Fig. 2 | Strategy for building α-helical backbone structure topologies.** Top: building blocks for generating backbone structures. Canonical α-helices and three types of HLH tertiary motifs typically observed in nature, hairpin (h), v-shaped (v) and corner (c), are used. Helices range from 5 to 35 residues, and each motif type comprises six patterns (Fig. 3a). The motif types were classified on the basis of the bending angle between the constituent helices in HLH motifs. Middle: secondary-structure element ordering to build α-helical proteins with five helices. According to the ordering, globular backbone structures without steric clashes are exhaustively explored by combining the building blocks, with the constraint of total residue length. Bottom: examples for generated α-helical backbone structure topologies. Poorly packed structures (lower) are discarded, whereas globularly folded structures (upper) are collected.

nearly all sides by multiple structural elements in three dimensions, like globins. Therefore, the ability to create protein structures with irregularly packed α-helices would contribute to the design of various functional proteins.

In this article, we sought to develop a computational method to design complicated all-α structures by employing helix–loop–helix (HLH) motifs typically observed in naturally occurring proteins. The developed method enabled us to generate a wide range of α-helical protein structure topologies from bundle-like to complicated by combining the typical HLH motifs and canonical α-helices. Finally, we demonstrated the ability to create complicated all-α proteins by de novo design of five distinct topologies.

## Strategy for all-α topology building

Although it has been suggested that the overall tertiary arrangements of helices of naturally occurring α-helical proteins can be approximated by a quasi-spherical polyhedral model[28], the major obstacle in designing complicated all-α topologies with irregularly aligned α-helices is attributed to the difficulty in determining a priori feasible topologies with their backbone blueprints involving lengths of secondary structures and loops. This is different from the design of αβ-proteins: the topologies are selected in advance by β-strand arrangements (that is, the order and orientations of β-strands in a β-sheet), and the backbone blueprints were derived from a set of rules relating local backbone structures of a few successive secondary structure elements to the

preferred tertiary motifs[29]. Therefore, we attempted to explore all-α topologies, not by preparing them a priori but by generating backbone structure topologies through the combinatorial enumeration of tertiary building blocks (Fig. 2). Moreover, the tertiary building blocks were selected from those typically observed in nature, so that the generated backbone structures are likely to be feasible. Therefore, the question is whether complicated all-α topologies can be generated from typical building blocks.

## A typical set of HLH motifs as building blocks

We first attempted to collect a set of HLH tertiary motifs that are typically observed in nature as building blocks. The HLH units consisting of two α-helices and the connecting loop of one to five residues in length were extracted from naturally occurring proteins, then clustered into 18 subgroups based on the five-dimensional feature vectors representing the HLH tertiary geometries[30] (Extended Data Figs. 2 and 3 and Methods). The representative 18 HLH motifs corresponding to each cluster density peak exhibited a broad range of bending angles between two helices, such as left- or right-handed helix–turn–helix, helix–corner–helix and kinked helices (Fig. 3a and Extended Data Figs. 3 and 4); the amino acid preference for each motif is shown in Extended Data Fig. 5: Gly at the residues with positive phi backbone torsion angle, helix capping residues immediately before helices such as Asp, Asn, Thr or Ser (refs. 31,32), and hydrophobic periodicity of helix residues specific to each motif are observed. The 18 HLH motifs are classified into three classes according to the magnitude of the bending angle: hairpin (h), v-shaped (v) and corner (c). The 18 representative HLH motifs were used as building blocks (Fig. 2, top) for generating α-helical backbone structure topologies.

## Generation of all-α topologies by combinatorial enumeration

Next, we investigated whether complicated topologies are produced using these typical tertiary motifs. Helical backbone structures composed of five and six helices were built with 90 and 110 residues in the total length, respectively, by combining the set of 18 HLH motifs and canonical α-helices ranging from 5 to 35 residues. The backbone structures were generated by enumerating all the combinations and selecting compact and steric-clash-free structures (Methods): 1,159,937,910 five-helix and 20,878,882,380 six-helix structures were enumerated, and 1,899,355 and 380,869 structures were then selected for each. The resulting topologies exhibited a broad spectrum ranging from helical bundle-like to complicated globular structures, demonstrating that complicated α-helical topologies are created from the typical tertiary motifs and canonical α-helices (Fig. 3b, white bar; Fig. 3c and Extended Data Fig. 6); the helix lengths were also widely distributed in the generated structures (Extended Data Fig. 7). Moreover, we found that the complexities of the generated topologies increase, as tertiary motifs with larger bending angles are included (black, gray and white bars in Fig. 3b). These results highlight the importance of corner-type motifs[33] in building complicated α-helical topologies.

## Design of complicated α-helical topologies

From the generated myriad backbone structure topologies, we selected five for de novo design, H5_fold-0, H6_fold-C, H6_fold-Z, H6_fold-U and H7_fold-K (the Arabic numeral after 'H' indicates the number of helices) (Fig. 4 and Supplementary Fig. 1), in the following way. We first selected three topologies exhibiting extremely low helix order (HO) values (for the definition, see Fig. 1c and Methods): H5_fold-0, H6_fold-C and H6_fold-Z (Fig. 1d). Next, to test whether all identified HLH motifs could be used for de novo design, we selected H6_fold-U and H7_fold-K, which include all of the HLH motifs not used in the first three and still exhibit lower HO values (Fig. 1d). For all target folds except H5_fold-0, the lengths of the terminal helices were manually elongated to ensure sufficient packing interactions. None of these

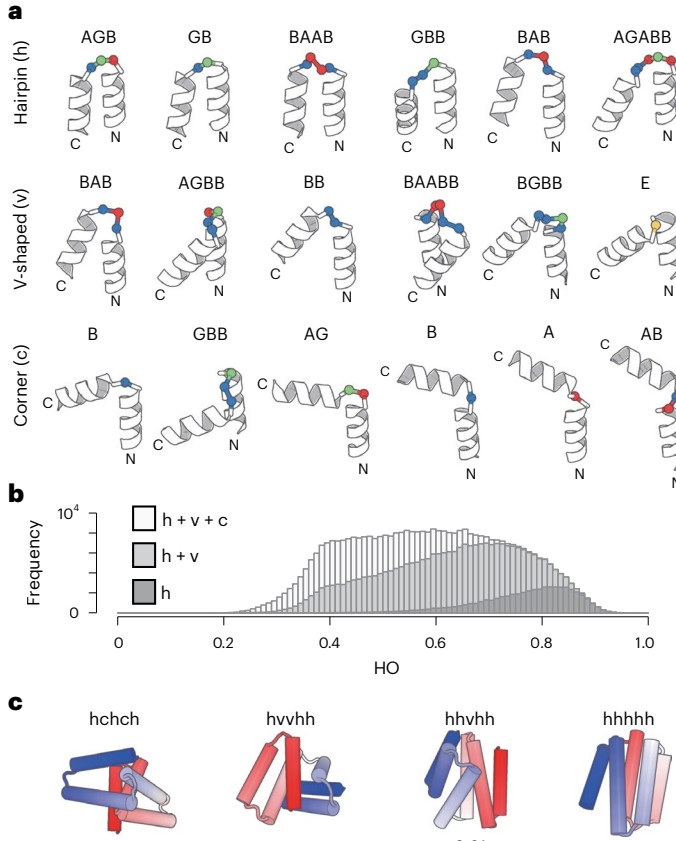

**Fig. 3 | 18 HLH tertiary motifs and generated α-helical backbone structures.**
**a**, Identified 18 HLH tertiary motifs typically observed in nature. The motifs are classified based on the bending angle between the two helices in the motifs: hairpin (h), v-shaped (v) and corner (c), which are presented in order of the magnitude of the bending angle, with the ABEGO backbone torsion pattern for the connecting loop. The residues with the backbone torsion angle, A, B, E and G, in the ABEGO torsion representation ('A' corresponds to the right-handed α-helix region in the Ramachandran map, 'B' to the β-strand region, 'E' to the extended region with a positive phi angle, and 'G' to a left-handed α-helix) are shown in red, blue, yellow and green, respectively. **b**, HO distributions for generated backbone structures with six helices. The black, gray and white bars respectively represent the distributions for the ensemble generated using only hairpin motifs (h), hairpin and v-shaped (h + v) motifs, and all three motifs (h + v + c). Incorporation of v-shaped and corner loops yields lower HO structures. **c**, Examples for the generated backbone structures. The used motif type strings and the HO values are indicated for each structure. The N- and C-terminals are colored in blue and red, respectively.

backbone structures is similar to any known protein structures; H5_fold-0, H6_fold-C, H6_fold-Z and H6_fold-U show a TM-score <0.6, using TM-align[34] against the ECOD database[35], and H6_fold-K shows a score of 0.610, with a structure of e2bnlA1 (Extended Data Fig. 8). The details of the selected topologies are described in Supplementary Text. For each backbone structure, amino acid sequences were designed through iterations of fixed-backbone sequence optimization and fixed-sequence structure optimization using Rosetta design calculations[36,37]. Designs with low energy, tight core packing[38] and high compatibility between local sequences and structures[29] were selected, and their energy landscapes were explored by 10,000 independent Rosetta ab initio structure prediction simulations starting from an extended conformation[39]. Ninety-one percent (75 of 82 designs) for H5_fold-0, 45% (18 of 40 designs) for H6_fold-C, 68% (27 of 40 designs) for H6_fold-Z, 67% (60 of 90 designs) for H6_fold-U, and 40% (36 of 90 designs) for H7_fold-K, showed funnel-shaped energy landscapes. Among the designs

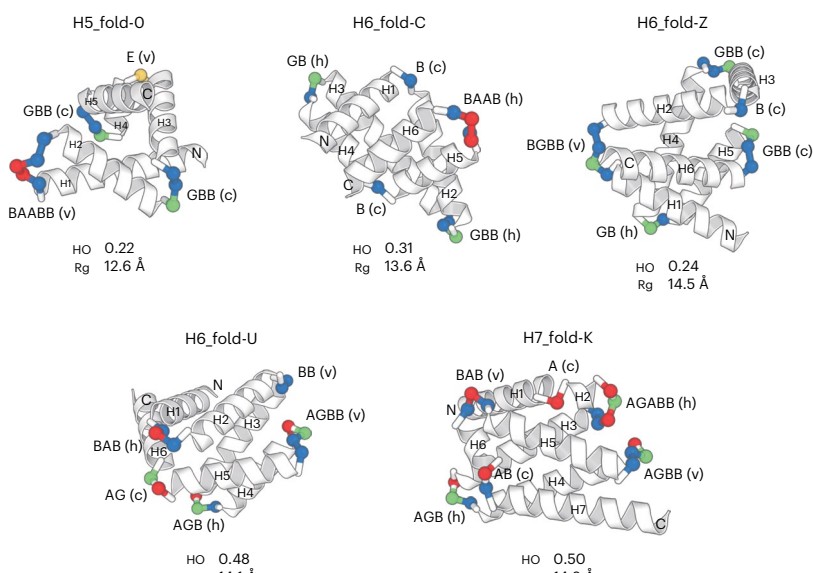

**Fig. 4 | Backbone structures for the five design target topologies.** The design target backbone structures. H1–7 represents the first to seventh helices. The letter string next to a loop indicates the ABEGO torsion pattern and the character within a bracket indicates the motif type. The loop residues are colored in the ABEGO torsion representation, same as Fig. 3a. The HO value and radius of gyration (Rg) are indicated for each structure.

## Table 1 | NMR constraints and structure statistics of the five designed structures

| Design ID | H5_fold-O_Chantal | H6_fold-C_Rei | H6_fold-Z_Gogy | H6_fold-U_Nomur | H7_fold-K _Mussoc |
|---|---|---|---|---|---|
| PDB ID | 7BQM | 7BQN | 7BQQ | 7BQS | 7BQR |
| BMRB entry | 36,335 | 36,336 | 36,337 | 36,339 | 36,338 |
| **NMR distance and dihedral constraints** | | | | | |
| Distance constraints | | | | | |
| Total NOE | 2,098 (100.0%) | 3,018 (100.0%) | 2,771 (100.0%) | 2,515 (100.0%) | 2,934 (100.0%) |
| Intra-residue | 425 (20.3%) | 596 (19.7%) | 551 (19.9%) | 436 (17.3%) | 484 (16.5%) |
| Inter-residue | | | | | |
| Sequential ($|i-j|$=1) | 555 (26.5%) | 727 (24.1%) | 620 (22.4%) | 578 (23.0%) | 661 (22.5%) |
| Medium range (1<$|i-j|$<5) | 617 (29.4%) | 983 (32.6%) | 867 (31.3%) | 801 (31.8%) | 906 (30.9%) |
| Long range ($|i-j|$≥5) | 501 (23.9%) | 712 (23.6%) | 733 (26.5%) | 700 (27.8%) | 883 (30.1%) |
| Total dihedral angle restraints | 132 | 206 | 232 | 195 | 220 |
| φ | 66 | 103 | 116 | 97 | 110 |
| ψ | 66 | 103 | 116 | 98 | 110 |
| **Structure statistics** | | | | | |
| Violations (mean and s.d.)† | | | | | |
| Distance constraints (Å) | 0.024±0.068 (0.10 ± 0.28) | 0.000±0.000 (0.00 ± 0.00) | 0.192±0.059 (1.20 ± 0.70) | 0.318±0.071 (1.50 ± 0.59) | 0.219±0.020 (1.20 ± 0.43) |
| Dihedral angle constraints (°) | 0.000±0.000 (0.00 ± 0.00) | 1.006±4.273 (0.05 ± 0.21) | 21.473±0.262 (0.90 ± 0.26) | 2.056±5.853 (0.10 ± 0.28) | 4.336±7.772 (0.20 ± 0.35) |
| Max. distance constraint violation (Å) | 0.243 | 0.186 | 0.413 | 0.416 | 0.278 |
| Max. dihedral angle violation (°) | 18.782 | 20.114 | 29.059 | 21.092 | 23.562 |
| Deviations from idealized geometry‡ | | | | | |
| Bond lengths (Å) | 0 | 0 | 0 | 0 | 0 |
| Bond angles (°) | 0 | 0 | 0 | 0 | 0 |
| Impropers (°) | 0 | 0 | 0 | 0 | 0 |
| Average pairwise RMSD* (Å) | | | | | |
| Heavy | 1.34±0.11 | 1.26±0.12 | 1.19±0.08 | 1.15±0.10 | 1.14±0.10 |
| Backbone | 0.39±0.09 | 0.47±0.10 | 0.39±0.06 | 0.35±0.06 | 0.27±0.05 |

†Mean and s.d. values are derived from 20 models of Amber refined structures. The averaged number of violations for dihedral angle constraints (>20°) and distance constraints (>0.2 Å) across 20 models, and the s.d. are indicated in parentheses. ‡No geometrical outliers are found in all models. *Averaged RMSD and deviation of backbone and heavy atoms for all pair of models in ensemble (20×20), are calculated by MolMol[53], fitted on the residues in ordered region (H5_fold-O_Chantal: 4–85; H6-fold-C_Rei: 2–15, 24–39, 69–91, 97–112; H6_fold-Z_Gogy: 4–118; H6_fold-U_Nomur: 5–19, 21–36, 38–96, 99–107; H7_fold-K_Mussoc: 8–25, 27–48, 52–121) identified by Filt_Robot[54].

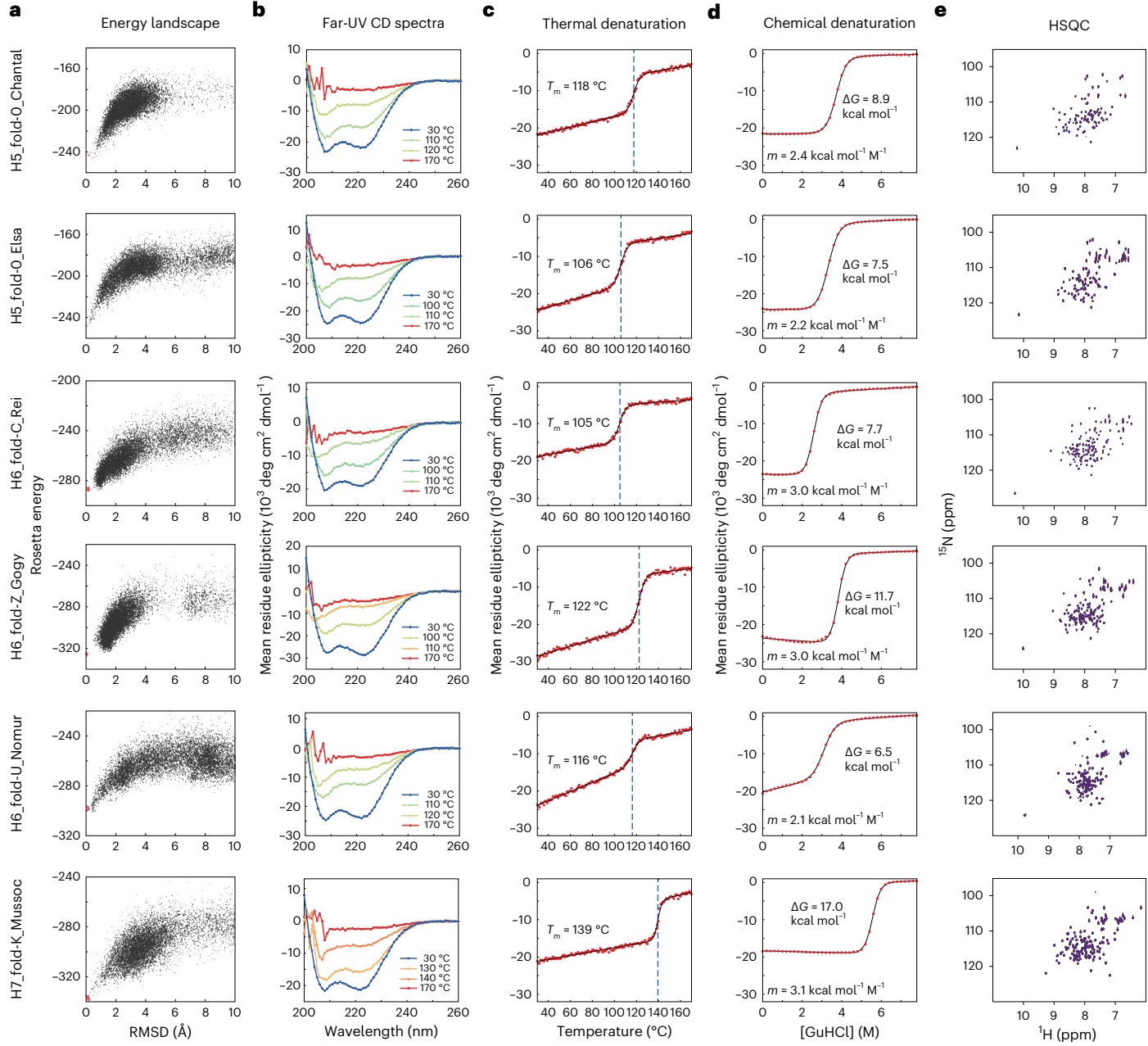

**Fig. 5 | Characterization of designed proteins. a**, Energy landscapes from Rosetta ab initio structure prediction simulations. The *y* axis represents Rosetta all-atom energy and the *x* axis represents the Cα RMSD from the design model. Black points represent the lowest energy structures obtained in independent Monte Carlo structure prediction trajectories starting from an extended chain for each sequence; red points represent the lowest energy structures obtained in trajectories starting from the design model. **b**, Far-ultraviolet CD spectra at 30 °C, the temperatures close to the melting temperature $T_m$, and 170 °C. The

CD spectra were recorded under the pressure of 10 bar. **c**, Thermal denaturation measured at 222 nm under the pressure of 10 bar. For each design, the data were fitted to a two-state model (black solid line) to obtain the $T_m$. **d**, Chemical denaturation with GuHCl (square brackets denote concentration) measured at 222 nm and 25 °C. For each design, the data were fitted to a two-state model (black solid line) to obtain the free energy of unfolding $\Delta G$ and its dependency on the denaturant, *m*-value. **e**, Two-dimensional $^1H$-$^{15}N$ HSQC spectra at 25 °C and 600 MHz.

having funnel-shaped energy landscapes, we selected approximately ten designs for each topology (for the details, see Methods).

## Experimental characterization of designed proteins

We obtained synthetic genes encoding ten designs for H5_fold-O, seven for H6_fold-C, seven for H6_fold-Z, eight for H6_fold-U and eight for H6_fold-K. Some designs (H6_fold-Z, 2; H6_fold-U, 1; H7_fold-K, 2) have weak sequence similarity to known proteins with blast *E*-value <0.005,

but the structures are unknown (Supplementary Table 1). The proteins were expressed in *Escherichia coli* and purified using a Ni$^{2+}$-NTA affinity column. The purified proteins were then characterized by circular dichroism (CD) spectroscopy and size-exclusion chromatography combined with multi-angle light scattering (SEC–MALS). For all design target topologies, 34 of 40 designed proteins were found to be well expressed and highly soluble, and showed CD spectra typical of α-helical proteins; 27 out of the 34 designs were found to be monomeric by SEC–MALS (Supplementary Tables 2–6). Furthermore, the monomeric designs

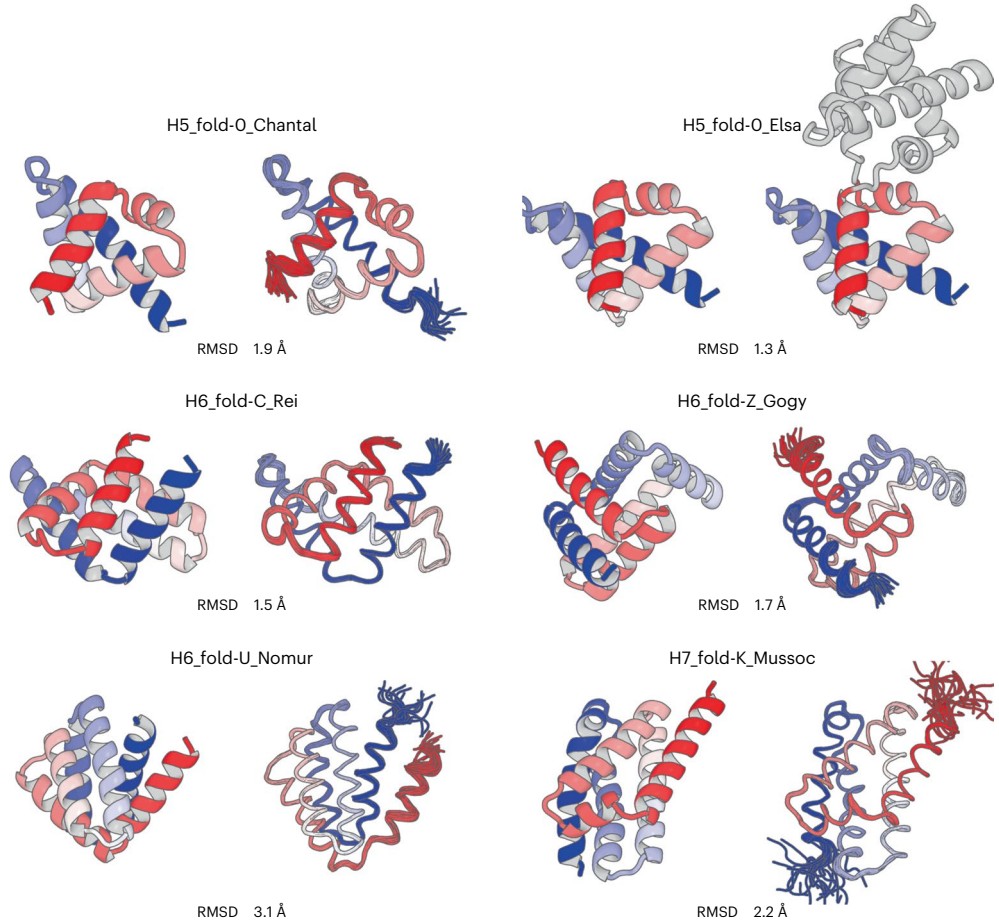

**Fig. 6 | Comparison of computational models with experimentally determined structures.** Design structures (left) and NMR structures (a crystal structure for H5_fold-0_Elsa) (right); the N- and C-terminals are colored in blue and red, respectively. The Cα RMSD between them is indicated (for H5_fold-0_Elsa, of which crystal structure is domain-swapped dimer, the Cα RMSD was calculated using MICAN[56]).

were characterized by $^1$H-$^{15}$N heteronuclear single quantum coherence (HSQC) nuclear magnetic resonance (NMR) spectroscopy, and 23 designs showed well-dispersed sharp peaks (Supplementary Tables 2–6 and Supplementary Fig. 2). The experimental results for all the designs are summarized in Extended Data Table 1. For each topology, we selected one monomeric design with well-dispersed sharp NMR peaks for NMR structure determination (Fig. 5 and Supplementary Fig. 3). All the designs were found to be highly stable from thermal denaturation up to 170 °C by CD (Fig. 5b,c). The NMR structures were solved at high quality using MagRO-NMRViewJ[40,41] (Table 1, Supplementary Text, Supplementary Figs. 4 and 5 and Supplementary Table 7), and the solved structures were consistent with the design models (Fig. 6 and Supplementary Table 8). For H5_fold-0, one of the designs was solved by X-ray crystallography and was nearly identical to the design model except for the domain swapping in the crystallized condition (Fig. 6, Table 2 and Supplementary Fig. 5). Despite the inclusion of noncanonical helix–helix packing arrangements in each design, the sidechains from distant α-helices were found to be coherently packed to constitute a single hydrophobic core similar to the design model. Notably, the bulky hydrophobic sidechains from the loops and neighboring α-helices also contributed largely to the core: they spiked the core and pinned the loops to the target conformations (Extended Data Fig. 9; for the importance of hydrophobic residues in the HLH motifs on energy landscapes of the designs, see Supplementary Fig. 6b,e). Interestingly, the N- and C-terminal helices of H6-FoldU_Nomur was found to be fluctuated despite the helix formation (Supplementary Figs. 7–9). Furthermore, in the thermal denaturation, the helical content of H6-FoldU_Nomur was gradually decreased before the transition (the second from the bottom in Fig. 5c), and in the chemical denaturation, the $m$-value, which represents the cooperativity, was lower than those of the other designs (Fig. 5d; note that $m$-values also depend on protein size, with larger proteins having larger $m$-values[42]; therefore, the H5_fold-0_Elsa and Chantal, which are smaller in size than the other designs, show lower $m$-values). These results would be attributed to the low hydrophobicity for the core-forming residues of the C-terminus: almost all of the residues are Ala (Supplementary Fig. 8). We also compared the loop geometries of all HLH motifs at the ABEGO level in the design models and experimental structures (Supplementary Fig. 10 and Supplementary Table 9) (for the importance of helix capping residues in the HLH motifs on energy landscapes of the designs, see also Supplementary Fig. 6c,f). Except for the loop immediately before the C-terminal helix of H6-FoldU_Nomur, all loop geometries of the experimental structures agreed with those of the design models. These results indicate that the difficult-to-describe α-helical proteins are designable with typical building blocks.

## Discussion
De novo designs of α-helical proteins have focused on structures consisting of parallelly aligned α-helices (Fig. 1), many of which are based on helical structure models such as the helical wheel[43] and Crick's parameterization[44]. We sought to develop a computational method for designing difficult-to-describe α-helical protein structures. We first identified the 18 HLH motifs typically observed in naturally occurring proteins. We then demonstrated that a wide range of globular all-α backbone structure topologies from bundle-like to complicated are generated by

**Table 2 | X-ray crystallography data collection and refinement statistics**

| | H5_fold-0_Elsa |
|---|---|
| **Data collection** | |
| Space group | $P2_1$ |
| Cell dimensions | |
| $a, b, c$ (Å) | 45.98, 33.66, 58.38 |
| $α, β, γ$ (°) | 90.00, 93.11, 90.00 |
| Resolution (Å) | 45.9–2.33 (2.47–2.33)* |
| $R_{merge}$ | 0.090 (0.483) |
| $I/σI$ | 13.63 (2.54) |
| Completeness (%) | 98.7 (92.3) |
| Redundancy | 6.4 (4.5) |
| **Refinement** | |
| Resolution (Å) | 45.9–2.33 |
| No. reflections | 49,835 |
| $R_{work}/R_{free}$ | 0.2066/0.2469 |
| No. atoms | |
| Protein | 1,422 |
| Ligand/ion | 6 |
| Water | 16 |
| *B*-factors | |
| Protein | 57.59 |
| Ligand/ion | 84.02 |
| Water | 45.80 |
| RMSD | |
| Bond lengths (Å) | 0.003 |
| Bond angles (°) | 0.517 |

*Statistics for the highest resolution shell are shown in parentheses. PDB ID: 7DNS.

combining the 18 typical HLH motifs and canonical α-helices. The key to building complicated α-helical topologies is to include HLH motifs with larger bending angles such as corner-type motifs. The approach of this developed method is regarded as the reverse of blueprint-based design: design target topologies are searched by the combinations of HLH motifs in this approach, whereas design target topologies are predetermined and then local backbone structures favoring the topologies are selected in blueprint-based design.

We succeeded in designing complicated α-helical protein structures with five distinct topologies, three of which, H5_fold-0, H6_fold-C and H6_fold-Z, exhibited structural complexities comparable to the globin fold. The design success rate was as high as that of previous de novo designs, and the design exhibited high solubility and thermal stability, similarly to previous designs[29,45–49]. Moreover, the loop geometries of almost all HLH motifs were formed as designed, which must have enabled the designed proteins to fold into the target topologies. These de novo design results indicate that the compact and steric-clash free backbone structures generated by using the typical HLH motifs are probably designable. In this regard, however, one of the questions is whether all or how much of the generated backbone structures can have tight core packing of sidechains. We have demonstrated that the selected five backbone structures are packable through de novo design, but the packability for the other backbone structures has not been clarified, which should be addressed in next works.

The computationally generated myriad of complicated all-α structures should provide diverse and heterogeneous molecular surfaces for engineering functions such as binding, enzymatic activity and self-assembly into symmetric oligomers. The myriad of generated structures, which are presumably highly soluble and stable, coupled with the recently developed massive gene synthesis[50,51] and parallel high-throughput screening[17,18,26,52], should make it possible to create proteins with optimal structures for specific functions[17,26].

## Online content

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

## Methods

### Definition of HO

HO is the order parameter that captures the complexities of α-helical proteins. HO is defined by the average of inner products between helix orientation vectors, $u_i$, for all pairs of $N$ α-helices[55]:

$$\text{HO} = \frac{2}{N(N-1)} \sum_{\substack{i=1 \\ i<j}}^{N} (\mathbf{u_i} \cdot \mathbf{u_j})^2.$$

Higher values indicate more ordered and lower values, more complicated.

### Analysis of all-α protein structures for de novo designed and naturally occurring proteins

Twenty-two de novo designed all-α protein structures were collected from Protein Data Bank (PDB). To this end, de novo designs were searched by the keyword 'de novo' or 'de-novo' in PDB as of November 2020, and then all-α structures containing no β-strands were extracted on the basis of the secondary structure assignments by the DSSP algorithm[57] (for the PDB structures including multiple chains or NMR models, the first chain or model was used). The following four classes of de novo designed proteins were excluded from the dataset: (1) designed proteins created on the basis of backbone structures of naturally occurring proteins, and those with sequence similarity higher than 0.90 (as an exception, the three-helix bundle structure designs (PDB code: 6DS9 and 2A3D) were both included because of their structural dissimilarity); (2) assemblies composed of one or two α-helices (for example, 3R3K and 1U7J); (3) repetitive structures such as α-solenoids (for example, 1MJ0 and 5K7V); (4) membrane proteins.

For naturally occurring all-α protein structures, 7,352 representative structures found in the mainly-α class in the CATH database[58] with sequence identity less than 40% were used.

For calculating the HO values of the collected structures, secondary structure elements and loops were assigned by DSSP[57] (α-helices are defined for the residue regions composed of at least five successive residues assigned as 'H' by the DSSP calculation). Note that the secondary structure assignments by DSSP are not always consistent with those originally defined by the authors. For example, the number of α-helices in the structures (PDB codes: 4TQL and 1P68) respectively designed with three and four α-helices were defined as four and five due to partially distorted α-helices.

### Clustering of HLH units using the five features representing a HLH geometry

A total of 13,667 HLH structures were extracted from 7,280 X-ray structures (secondary structures were assigned by DSSP[57]), obtained from the PISCES server[59], with resolution ≤2.5 Å, R-factor ≤0.3, sequence length more than or equal to 40, and ≤25% sequence identity. We then classified the HLH structures by their loop residue lengths and extracted 13,510 HLH structures in total with loop of one to five residues in length. The extracted HLH structures were clustered for each loop length from one to five using the density clustering algorithm[30] (Extended Data Fig. 3), with the five features representing a HLH geometry (Extended Data Fig. 2).

### Building backbone structures

α-Helical backbone structures were built using Rosetta by exhaustive sampling for the conformations with steric-clash free (Rosetta vdw score <4.0 using the weight value, 0.1) and smaller radius of gyration (<14 Å) (the threshold value corresponds to the peak of the distribution of the radius of gyration for naturally occurring proteins; Supplementary Fig. 11) by combining canonical α-helices ranging from 5 to 35 residues (backbone torsion angles, phi, psi and omega, were set to −60.0, −45.0 and 180.0, respectively) and the identified 18 HLH motifs (Main and Fig. 3a), with length constraints of 90 and 110 residues for the five- and six-helix proteins, respectively. For generating five-helix structures, 64,440,995 steric-clash free four-helix structures with 70 residues were first generated, and then an α-helix with 18 types of connecting loops was appended to the C-terminal of the generated four-helix structures so that the total length becomes 90 residues. For generating six-helix structures, an α-helix with 18 types of connecting loops was appended to the N-terminal of the generated five-helix structures so that the total length becomes 110 residues. From these structures, the globular five- and six-helix structures were collected on the basis of the radius of gyration.

### Selection of designs for experimental characterization based on the shapes of energy landscapes

We set three criteria for the selection by the shapes of energy landscapes. First, the overall shape of the landscape should be funnel-like with an apparent and sharp 'neck' reaching low-root mean square deviation (RMSD) and low-energy region, which is the hallmark of the foldability specifically into the target conformation. This is the most important criterion on the selection of energy landscape plots: for ill-designed sequences, all conformations remain in the high-RMSD and high-energy regions and do not have such a 'neck'. Second, the funnel should not have subminima that indicate that the protein has alternative folded states. This is a criterion to exclude the possibility of misfolding and avoid a rugged energy landscape. Third, the ensemble of lowest-RMSD and lowest-energy conformations at the bottom of the funnels should not be away from, and ideally should overlap with, the conformational ensemble in the simulations starting from the target structure. This criterion is not mandatory, but consistency between fragment assembly simulations that offer global sampling and near-native relax simulations helps us to rank the designs with the similar quality in terms of the first and second criteria.

### Expression and purification of designed proteins

The genes encoding the designed sequences were synthesized and inserted into pET21b vectors. The whole plasmid constructs were purchased from FASMAC or Eurofins Genomics. The target proteins were overexpressed by IPTG induction in *E. coli* BL21 Star (DE3) cells cultured in MJ9 minimal media including $^{15}N$ ammonium sulfate as the sole nitrogen source and $^{12}C$ glucose as the sole carbon source[60]. The expressed uniformly (*U*-)$^{15}N$-labeled proteins with a 6xHis tag at the C-terminus were purified by $Ni^{2+}$-affinity columns. The purified proteins were then dialyzed against phosphate-buffered saline (PBS) buffer, 137 mM NaCl, 2.7 mM KCl, 10 mM $Na_2HPO_4$ and 1.8 mM $KH_2PO_4$, at pH 7.4; this buffer was used for all the experiments except NMR structure determination. The expression level, solubility and purity of each designed protein were evaluated by sodium dodecyl sulfate–polyacrylamide gel electrophoresis. To further confirm them, the samples were analyzed by mass spectroscopy (Bruker Daltonics REFLEX III and Thermo Scientific Orbitrap Elite).

### Experiments to identify designed proteins exhibiting folding ability

The following three experiments were conducted to evaluate the folding ability of designed sequences: CD spectroscopy, size exclusion chromatography with multi-angle light scattering (SEC–MALS) and $^1H$-$^{15}N$ HSQC NMR spectroscopy. Supplementary Tables 2–6 present the results of the evaluations for each designed sequence for each fold.

### CD spectroscopy under 1-bar pressure

Far-UV CD spectra was measured to study whether the designs show the characteristic spectra of α-helical proteins, by scanning from 260 to 200 nm at 20 °C for ~15 μM protein samples in PBS buffer on a JASCO

J-1500 CD spectrometer. The measurements were performed four times and then averaged.

## SEC−MALS

Oligomeric states for the designs in solution were studied by SEC−MALS with miniDAWN TREOS static light scattering detector (Wyatt Technology Corp.) combined with a high-performance liquid chromatography system (1260 Infinity LC, Agilent Technologies) with a Shodex KW-802.5 column (Showa Denko K.K.) for H5_fold-0_Chantal and H6_fold-C_Rei or a Superdex 75 increase 10/300 GL column (GE Healthcare) for H5_fold-0_Elsa, H6_fold-Z_Gogy and H7_fold-K_Mussoc. After the equilibration of the column with PBS buffer, 100 μl of the samples after purification by Ni$^{2+}$-affinity columns were injected. The absorbance at 280 nm was measured by the high-performance liquid chromatography system to give the protein concentrations and intensity of light scattering at 659 nm was measured at angles of 43.6°, 90.0° and 136.4°. These data were analyzed by the ASTRA software (version 6.1.2, Wyatt Technology) using a change in the refractive index with concentration, a d$n$/d$c$ value, 0.185 ml g$^{-1}$, to estimate the molecular weight of dominant peaks.

## $^1$H-$^{15}$N HSQC NMR spectroscopy

Whether the designs fold into well-packed structures or not was evaluated by $^1$H-$^{15}$N HSQC 2D-NMR spectroscopy. The purified protein samples were concentrated to 0.2–1.0 mM, and mixed with their 10% volume of D$_2$O. The experiments were performed at 25 °C on a JEOL JNM-ECA 600 MHz spectrometer, and data were analyzed by JEOL Delta (version 5.3.1).

## High-pressure CD spectroscopy for melting temperature ($T_m$) estimation

For the designs that were evaluated to have the folding ability in the above experiments (one design for each target topology was selected), thermal denaturation was studied by using high-pressure CD spectroscopy. JASCO J-1500 CD spectrometer was equipped with additional pressure instruments so that temperature of the solution samples can be scanned from 30 °C to 170 °C under 10 bar. Temperature was increased 1 °C per minute for ~15 μM protein samples. Fixed wavelength measurements at 222 nm were performed at every 1 °C, and wavelength scanning measurements (260 to 200 nm) were performed at 30, 40, 60, 80, 90, 100, 110, 120, 130, 140, 150, 160 and 170 °C. Thermal denaturation was measured once. $T_m$ was estimated by nonlinear fitting to thermal denaturation CD curve at 222 nm. The nonlinear least-squares analysis was performed by nls function in R language, given a two-state unfolding and linear extrapolation model. After this fitting, we obtained $T_m$ at which the estimated populations of folded and unfolded states become equal.

## CD spectroscopy for chemical denaturation

Chemical denaturation with GuHCl was monitored at 222 nm for 2–3 μM protein samples in PBS buffer (pH 7.4) at 25 °C in a 1-cm path length cuvette. The GuHCl concentration was automatically controlled by a JASCO ATS-530 titrator. Chemical denaturation was measured once. The chemical denaturation curves were fit by nonlinear least-squares analysis using a two-state unfolding and linear extrapolation model[61]. The free energy change, Δ$G$, for the unfolding transition and its dependency on the denaturant, $m$-value, were obtained from the fitting.

## Sample preparation for NMR structure determination

The most promising design for each target topology was overexpressed by IPTG induction in *E. coli* BL21 Star (DE3) cells cultured in MJ9 minimal media containing $^{15}$N ammonium sulfate as the sole nitrogen source and $^{13}$C glucose as the sole carbon source[60]. The expressed $U$-$^{15}$N,$U$-$^{13}$C-enriched proteins were purified by Ni$^{2+}$-affinity columns, and dialyzed against PBS buffer. The protein samples were further purified by gel filtration chromatography on an ÄKTA Pure 25 FPLC (GE Healthcare) using a Superdex75 or Superdex75 increase 10/300 GL column (GE Healthcare), which also replaced the PBS buffer at pH 7.4 with the customized buffer for NMR spectroscopy. The following 95% H$_2$O/5% D$_2$O buffer conditions for each sample were used: 100 mM NaCl, 5.6 mM Na$_2$HPO$_4$, 1.1 mM KH$_2$PO$_4$, at pH 7.4 for H5_fold-0_Chantal; 50 mM NaCl, 5.5 mM Na$_2$HPO$_4$, 4.5 mM KH$_2$PO$_4$, at pH 6.9 for H6_fold-C_Rei; 50 mM NaCl, 3.2 mM Na$_2$HPO$_4$, 4.5 mM KH$_2$PO$_4$, at pH 6.5 for H6_fold-Z_Gogy; 155 mM NaCl, 3.0 mM Na$_2$HPO$_4$, 1.1 mM KH$_2$PO$_4$, 10 μM ethylenediaminetetraacetic acid, 0.02% NaN$_3$, cOmplete protease inhibitor cocktail (Roche), at pH 7.4 for H6_fold-U_Nomur; and 155 mM NaCl, 3.0 mM Na$_2$HPO$_4$, 1.1 mM KH$_2$PO$_4$, at pH 7.4 for H7_fold-K_Mussoc.

## Solution structure determination by NMR

**NMR measurements.** NMR measurements were performed on Bruker AVANCE III NMR spectrometers equipped with QCI cryo-Probes at 303 K. The spectrometers with 600, 700 and 800 MHz magnets were used for the signal assignments and nuclear Overhauser effect (NOE)-related measurements, while 700, 900 and 950 MHz ones, for residual dipolar coupling (RDC) experiments. For the signal assignments, 2D $^1$H-$^{15}$N HSQC (echo/anti-echo), $^1$H-$^{13}$C Constant-Time HSQC for aliphatic and aromatic signals, 3D HNCO, HN(CO)CACB and 3D HNCACB for backbone signal assignments, while BEST pulse sequence was applied to the triple resonance measurements for H6_fold-C_Rei. For structure determination, 3D $^{15}$N-edited NOESY and 3D $^{13}$C-edited NOESY for aliphatic and aromatic signals (mixing time 100 ms) were performed. For H6_fold-U_Nomur, additional 3D HN(CA)CO, HN(CO)CA, HNCA, HBHA(CO)NH, HBHANH, H(CCCO)NH, CC(CO) NH, 3D $^{13}$C-HSQC ($^{13}$C-t1) NOESY $^{13}$C-HSQC, 3D $^{13}$C-HSQC ($^{13}$C-t1) NOESY $^{15}$N-HSQC and 4D $^{13}$C-HSQC NOESY $^{13}$C-HSQC were measured. Except for 3D-edited NOESY, all the other spectra were performed using non-uniform sampling (NUS) for H6_fold-U_Nomur and H7_fold-K_Mussoc. For NUS, sampling ratio was set at 25% for 3D and 6% for 4D with a fixed random seed. The NUS spectra were reconstructed by iteratively re-weighted least squares for 3D while iterative soft thresholding for 4D spectra with virtual-echo technique using qMDD tool[62].

For the RDC experiments, 2D in-phase and anti-phase (IPAP) $^1$H-$^{15}$N HSQC using water-gate pulses for water suppression were measured with or without 6–10 mg ml$^{-1}$ of Pf1 phage (ASLA Biotech). For confirming the positions of $^1$H-$^{15}$N signals in the 2D IPAP $^1$H-$^{15}$N HSQC, 3D HNCO at the identical buffer condition containing Pf1 phage were measured. The α- and β-states of $^{15}$N signals split by $^1$H-$^{15}$N $^1$J-coupling were separately identified for the protein in the isotropic and weakly aligned states, to obtain 1-bond RDC $^1D_{1_{H/15N}}$ values. For the sample H6_fold-U_Nomur, 3D J-HNCO (without $^1$H decoupling for $^{15}$N evolution) was measured at 25% NUS, which were used for confirming α- and β-states of $^{15}$N signal positions overlapped in 2D IPAP spectra. 3D J-HN(CO)CA spectrum was also measured for H6_fold-U_Nomur to obtain $^1D_{1_{Hα/13Cα}}$ for appending an additional number of alignment data at the identical magnetic field and alignment tensor.

**NMR signal assignments.** All NMR signals were identified in a fully automated manner using MagRO-NMRViewJ (upgraded version of Kujira[40]), in which noise peaks were filtered by deep-learning methods using Filt_Robot[41]. FLYA module was used for fully automated signal assignments and structure calculation[63] to obtain roughly assigned chemical shifts (Acs), and then trustworthy ones were selected into the MagRO Acs table. After confirmation and correction of the Acs by visual inspection using MagRO, TALOS+[64] calculations were performed to predict phi/psi dihedral angles, which were then converted to angle constraints for the CYANA format.

**Structure calculation.** Several CYANA[65] calculations were performed using the Acs table, NOE peak table and dihedral angle constraints. The Acs table was exported by the MagRO CYANA module, and then the

aliased chemical shifts were automatically calculated depending on the spectrum width of responsible NOESY spectra. For dihedral angle constraints, phi and psi, with deviation were derived from TALOS+ prediction using chemical shifts of $^{15}N$, $^{13}C'$, $^{13}C\alpha$ and $^{13}C\beta$, with high prediction score noted by 'Good'. The minimal angle deviation was set at 20°. After several iterations of CYANA calculations, dihedral angle constraints derived from TALOS+[64] revealing large violation for nearly all models in structure ensemble were eliminated.

After the averaged target function of the ensemble reached to less than 2.0 Å$^2$, refinement calculations by Amber12 were carried out for 20 models with lowest target functions. The coordinates of final.pdb calculated by CYANA, distance constraints (final.upl), dihedral angle constraints derived from TALOS+ prediction were converted into Amber format and topology file using Sander Tools. Firstly, 500 steps of minimization (250 steps of steepest decent, 250 steps of conjugate gradient) were carried out without electrostatic potential and NMR constraints. Second, molecular dynamics simulations with the ff99SB force field using implicit water system (0.1 M of ionic strength, 18.0 Å of cutoff) were performed, in which the temperature was gradually increased from 0.0 K to 300.0 K by 1,500 steps, followed by the simulation with 28,500 steps at 300.0 K (1.0 fs time step, total 30 ps). Finally, 2,000 steps for minimization (1,000 steps for steepest decent and 1,000 steps for conjugate gradient) with constraints of distance and dihedral angle were applied at the same condition used in the molecular dynamics simulations.

**NMR structure validation.** The RMSD values were calculated for the 20 structures overlaid to the mean coordinates for the ordered regions, automatically identified by Filt_Robot using multi-dimensional non-linear scaling[54].

The RDC back-calculation was performed by PALES[66] using experimentally determined values of RDC. The averaged correlation between the simulated and experimental values was obtained using the signals except the residues on overlapped regions in $^1H$-$^{15}N$ HSQC and the ones in low-order parameters less than 0.8 predicted by TALOS+. For the validation of H6_fold-U_Nomur, a lot of signals were overlapped in 2D IPAP-HSQC spectra. To overcome this problem, $^1J_{HN-^{15}N}$ split 3D HNCO (without $^1H$-decoupling scheme in $^{15}N$ evolution period) spectra in isotropic and anisotropic states were measured by NUS (25% data point reduction) to obtain signal positions of α- and β-states of $^{15}N$ spins at resolution of 0.3 Hz. $^1J_{H\alpha/^{13}C\alpha}$ split 3D HN(CO)CA spectra at the same conditions were also measured to obtain $^1D_{^1H\alpha/^{13}C\alpha}$ at resolution of 0.2 Hz. Initially the RDC reproducibility of H6_fold-U_Nomur were examined using separately $^1D_{HN-^{15}N}$ and $^1D_{H\alpha-^{13}C\alpha}$ tables by PALES for all models to confirm that the averaged correlation coefficients are greater than 0.9, and then final correlation coefficients were calculated with two merged tables.

**Solution structural dynamics of H6_fold-U_Nomur measured by NMR**
**$^{15}N$ $R_1$, $R_2$ and $^{15}N$-{$^1H$} NOE experiments.** The $^{15}N$ $R_1$, $R_2$ and $^{15}N$-{$^1H$} NOE measurements were performed for a uniformly $^{15}N$-labeled H6_fold-U_Nomur protein sample with a concentration of 0.78 mM, which is the same condition as the solution used for the structure determination. These were conducted at 303 K on Bruker 700 MHz Avance-III NMR spectrometer equipped with cryogenic probe, using the 4-mm-diameter NMR Shigemi-tube. The $^{15}N$ $R_1$ and $R_2$ were obtained by measuring 2D $^1H$-$^{15}N$ HSQC with the inversion-recovery technique and with the temperature-compensated CPMG method, respectively[67]. Steady-state $^{15}N$-{$^1H$} NOE was obtained by measuring 2D $^1H$-$^{15}N$ HSQC spectra with and without saturation pulse in each of the retardation time acquired by the interleaved method. The 2D $^1H$-$^{15}N$ peaks were automatically identified and assigned using the MagRO software[40]. Some assignments were corrected with visual inspection. The $^{15}N$-{$^1H$} NOE values were estimated as the peak intensity ratio $I/I_0$ derived from

the 2D HSQC spectra with ($I$) and without ($I_0$) saturation pulse. The $I/I_0$ data were fitted by using an exponential equation, $I/I_0 = \exp(-R \times t)$ with delay time $t$ (s) to obtain the $^{15}N$ relaxation rate constant $R$ (s$^{-1}$).

**2D $^1H$-$^{15}N$ CLEANEX-PM FHSQC experiments.** The uniformly $^{15}N$-labeled protein sample of H6_fold-U_Nomur was lyophilized, and then 2D $^1H$-$^{15}N$ HSQC data were collected immediately after dissolving the lyophilized sample in 100% D$_2$O. However, protons of the amide groups of most residues were promptly replaced by deuterium within 10 min after the dissolution, probably due to the high pH of the sample solvent (pH 7.4). This prevented us to obtain practical H−D exchange rates. Therefore, the exchange rates between the water and amide protons were obtained using the 2D $^1H$-$^{15}N$ CLEANEX-PM FHSQC[68,69] scheme. In this method, the exchange ratio depends only on $k_{open}$ in the protein folding/unfolding. The amide group would be in the EX1 limit due to the relatively high pH of 7.4, namely $k_{close} \ll k$, where $k_{close}$ is the global and/or local folding rate of a protein and $k$ is the exchange rate of amide group in the unfolded state, the observable solvent exchange rate $k_{ex}$ would be obtained as the global and/or local unfolding rate of a protein, $k_{open}$. The 2D $^1H$-$^{15}N$ FHSQC data without applying spin-lock pulse was also measured under the same condition to obtain the reference, $I_0$. For 2D $^1H$-$^{15}N$ CLEANEX-PM FHSQC spectra with different spin-lock time $t_m$ and the reference spectrum, the observed peaks were automatically identified and assigned by MagRO[40] with manual correction to obtain a normalized list of signal intensities for each residue. The following equation was used to obtain $k_{obs}$ for each residue:

$$\frac{I}{I_0} = \frac{k_{ex}}{k_{ex} + R_{1A} - R_{1B}} \times \left\{ \exp\left(-R_{1B} \times t_m\right) - \exp\left[-\left(R_{1A} + k_{ex}\right) \times t_m\right] \right\},$$

where $R_{1B}$ is the apparent longitudinal relaxation rate of water molecules, and $R_{1A}$ is a mixture of the apparent longitudinal and transverse relaxation rates on the rotational frame for the residue of interest. The values of $R_{1A}$ and $k_{ex}$ for each residue with error values were obtained by curve-fitting by this equation, with the assumption, $R_{1B} = 0.6$ (s$^{-1}$).

**X-ray structure determination of H5_fold-0_Elsa**
**Sample preparation for X-ray structure determination.** The gene encoding the designed sequence of H5_fold-0_Elsa in pET21b vector was digested at the NdeI and XhoI restriction sites and cloned into pET15b-TEV vector with cleavable sites by TEV protease instead of thrombin (original) between the designed sequence and the N-terminal 6xHis tag. Designed protein was expressed in *E. coli* BL21 Star (DE3) cells, and purified by a Ni$^{2+}$-affinity column. The N-terminal His tag was then cleaved by TEV protease, and removed through a Ni$^{2+}$-affinity column. The protein samples without a His tag were purified by an anion-exchange chromatography (HiTrapQ HP 1-ml column, GE Healthcare) followed by gel filtration chromatography (Superdex 75 10/300 GL column) on an ÄKTA Pure 25 FPLC. Mass spectroscopy was performed to confirm that a His tag was successfully cleaved.

To assess the effect of the tag cleavage on the oligomeric state and stability, we performed SEC−MALS and thermal denaturation CD experiments under high pressure for the original and tag-cleaved samples of H5_fold-0_Elsa. The solvent was exchanged to PBS at pH 7.4 before these experiments. The results showed that the tag-cleaved protein was also monomeric and had nearly identical denaturation temperature (the second row in Fig. 5c, 106 °C) as the original sample with the C-terminal His tag (Supplementary Fig. 12, 105 °C), which indicates that the removal of tag and slight differences in flanking amino-acid sequences do not largely change the stability and oligomeric state of the designed protein in solution.

**Crystallization and X-ray structure determination.** The protein samples of H5_fold-0_Elsa at the concentration of 12 mg ml$^{-1}$ (1.07 mM) was crystallized in the solution of 0.4 M MgCl$_2$, 0.1 M Tris−HCl (pH 7.5) and

# Article

30% PEG 3350, using the sitting-drop vapor diffusion method at 296 K. The obtained crystals were soaked in the solution of 0.4 M $MgCl_2$, 0.1 M Tris–HCl (pH 7.5), 30% PEG 3350 and 10% glycerol, mounted on cryo-loops (Hampton Research), flash-cooled and stored in liquid nitrogen.

X-ray diffraction data of the crystal were collected with BL-1A beamline ($\lambda = 1.1000$ Å) at Photon Factory, and processed to 2.3 Å by XDS[70]. After phase determination by molecular replacement using the design model by Molrep[71] in the CCP4 suite, the molecular model was constructed and refined using Coot[72] and Phenix Refine[73]. Translation/Libration/Screw refinement was performed in late stages of refinement. The refined structures were validated with RAMPAGE[74]. Ramachandran plot statistics showed that 98.8% and 0.00% of residues were in favored and outlier regions, respectively. The crystallographic data collection is summarized in Table 2.

### Reporting summary

Further information on research design is available in the Nature Portfolio Reporting Summary linked to this article.

### Data availability

The solution NMR structures of the five designs have been deposited in the PDB under the accession numbers 7BQM (H5_fold-0_Chantal), 7BQN (H6_fold-C_Rei), 7BQQ (H6_fold-Z_Gogy), 7BQS (H6_fold-U_Nomur) and 7BQR (H7_fold-K_Mussoc). The NMR data have been deposited in the Biological Magnetic Resonance Data Bank under the accession numbers 36335 (H5_fold-0_Chantal), 36336 (H6_fold-C_Rei), 36337 (H6_fold-Z_Gogy), 36339 (H6_fold-U_Nomur) and 36338 (H7_fold-K_Mussoc). The crystal structure of H5_fold-0_Elsa has been deposited in the PDB under the accession number 7DNS. The computational design models are presented as Supplementary Data 1. The generated compact and steric-clash-free five-helix (1,899,355) and six-helix (380,869) structures are available at https://github.com/kogalab21/all-alpha_design. The plasmids encoding the designed sequences are available through Addgene under the accession numbers 201825 (H5_fold-0_Elsa), 201826 (H5_fold-0_Chantal), 201827 (H6_fold-C_Rei), 201828 (H6_fold-Z_Gogy), 201829 (H6_fold-U_Nomur) and 201830 (H7_fold-K_Mussoc). Source data are provided with this paper.

### Code availability

The code for building and analyzing helical structures has been implemented into Rosetta at https://github.com/RosettaCommons/main/tree/koga/all-alpha_design. The demo for building helical structures is available at https://github.com/kogalab21/all-alpha_design.

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

### Acknowledgements

We thank RIKEN Yokohama NMR Facility for NMR measurements, the Functional Genomics Facility, NIBB Core Research Facilities, especially Y. Makino, for mass spectrometry analysis, and the Instrument Center, Okazaki, Japan, especially M. Nakano, for HSQC spectra measurements. We also thank M. Yamamoto for experimental assistance, S. Minami for advice on structure similarity comparison and useful discussions, and Y. Ishii and S. Akiyama for valuable comments on the paper. The computations were performed using the Research Center for Computational Science (RCCS), Okazaki, Japan (Project: 21-IMS-C174, 20-IMS-C157, 19-IMS-C175, 18-IMS-C155, 17-IMS-C147, 16-IMS-C129 and 15-IMS-C180). Three-dimensional structure determination was supported by Basis for Supporting Innovative Drug Discovery and Life Science Research (BINDS) from AMED under grant numbers JP19am0101072 and JP20am0101083. The synchrotron radiation experiments were performed at Photon Factory (proposals 2016G-048). We also thank the beamline staff at BL1A of Photon Factory (Tsukuba, Japan) for help during data collection. This work was supported by the Japan Society for the Promotion of Science (JSPS) KAKENHI Grants-in-Aid for Scientific Research 15H05592 to N. Koga, 18H05420 to T.K. and N. Koga, 18H05425 to T.M. and 18K06152 to Naohiro K., the Japan Science and Technology Agency (JST) Precursory Research for Embryonic Science and Technology (PRESTO, grant number JPMJPR13AD to N. Koga) and the JST-Mirai Program (JPMJMI17A2 to Naohiro K.). K. Sakuma was also supported by JSPS KAKENHI Grant-in-Aid for JSPS Research Fellow 15J02427.

### Author contributions

K. Sakuma and N. Koga designed the research. K. Sakuma analyzed natural proteins and performed computational

design work. K. Sakuma wrote the program code. K. Sakuma, T.K. and R.T.-K. expressed, purified, characterized the designed proteins by biochemical assay, and prepared protein samples for NMR: H5_fold-O, H6_fold-C and H6_fold-Z, by K. Sakuma, and H6_fold-U and H7_fold-K, by T.K. and R.T.-K. For NMR structure determination, Naohiro K., T.S. and T.N. collected data: H5_fold-O_Chantal, H5_fold-C_Rei and H6_fold-Z_Gogy, by T.S., and H6_fold-U_Nomur and H7_fold-K_Mussoc, by Naohiro K. and T.N. Naohiro K. performed structural analysis. For NMR structural dynamics of H6_fold-U_Nomur, Naohiro K. and T.N. collected and analyzed data. For crystal structure determination, Naoya K. prepared protein samples and K. Suzuki, with advice from T.M., performed crystallization and structural analysis. K. Sakuma, Naohiro K., T.F., K. Suzuki, T.M., T.K., R.T.-K. and N. Koga wrote the paper.

## Competing interests

The authors declare no competing interests.

## Additional information

**Extended data** is available for this paper at https://doi.org/10.1038/s41594-023-01147-9.

**Correspondence and requests for materials** should be addressed to Nobuyasu Koga.

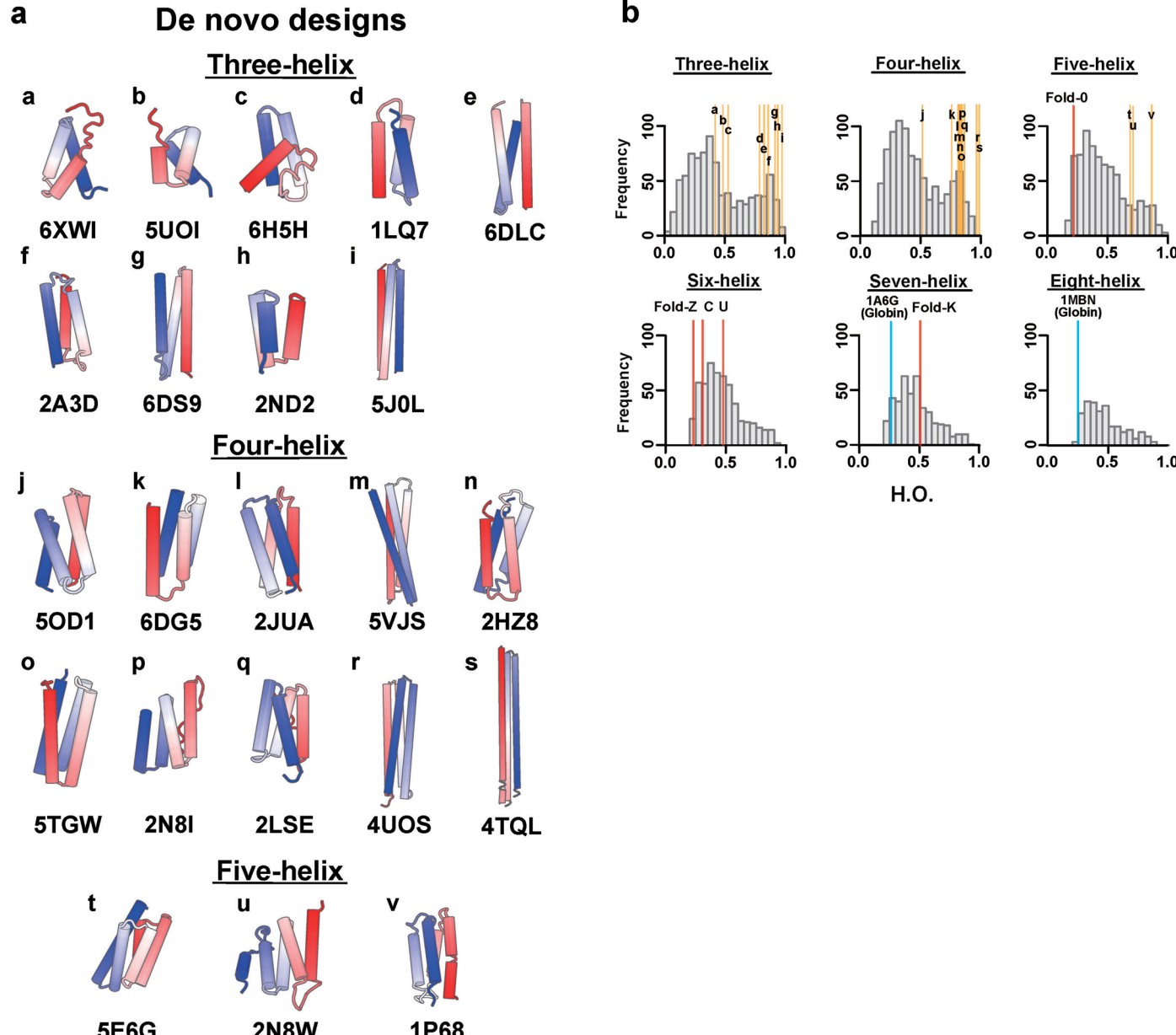

**Extended Data Fig. 1 | Twenty-two de novo designed proteins collected from Protein Data Bank (PDB). a**, Structures and their PDB IDs of the twenty-two de novo designed proteins collected from PDB. **b**, The helix order (HO) values of the designs were plotted in the HO histograms for naturally occurring all-α proteins (these histograms are identical to those in Fig. 1).

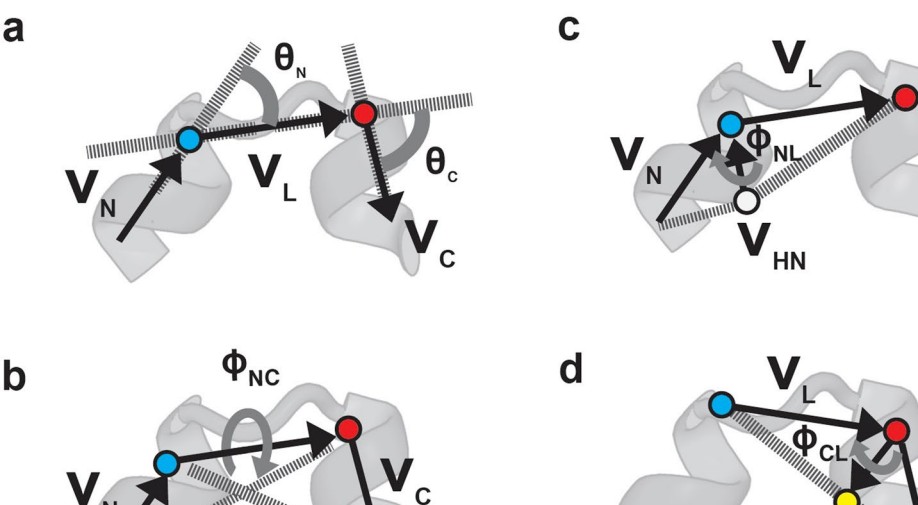

**Extended Data Fig. 2 | The five features representing the HLH tertiary geometry. a–d,** For representing tertiary geometries of HLH units, the following angles, $\theta_N$, $\theta_C$, $\phi_{NC}$, $\phi_{NL}$, and $\phi_{CL}$, were identified using the $\mathbf{V_N}$, $\mathbf{V_C}$, $\mathbf{V_L}$, $\mathbf{V_{HN}}$, and $\mathbf{V_{HC}}$ vectors (these vectors are calculated using Cα atoms). **a,** The definitions of $\theta_N$ and $\theta_C$. $\mathbf{V_N}$ and $\mathbf{V_C}$ respectively represent the helix vectors for the N- and C- terminal helices in a HLH geometry, which are calculated using the equations proposed by Krissinel et al.[53]. $\mathbf{V_L}$ is the loop vector from the last Cα atom (blue) in the N-terminal helix to the first Cα atom (red) in the C-terminal helix. $\theta_N$ was identified as the angle between the $\mathbf{V_N}$ and $\mathbf{V_L}$ vectors; $\theta_C$, was identified as the angle between the $\mathbf{V_C}$ and $\mathbf{V_L}$ vectors. **b,** The definitions of $\phi_{NC}$. $\phi_{NC}$ was identified as the dihedral angle between the plane defined with the $\mathbf{V_N}$ and $\mathbf{V_L}$ vectors and

that with the $\mathbf{V_C}$ and $\mathbf{V_L}$ vectors. **c,** The definition of $\phi_{NL}$. $\mathbf{V_{HN}}$ is the helix spiral vector at the end of the N-terminal helix, which was identified as the vector pointed to the last Cα atom (blue) in the N-terminal helix from the Cα atom immediately before the last Cα atom (white). $\phi_{NL}$ was identified as the dihedral angle between the plane defined with the $\mathbf{V_{HN}}$ and $\mathbf{V_L}$ vectors and that with the $\mathbf{V_{HN}}$ and $\mathbf{V_N}$ vectors. **d,** The definition of $\phi_{CL}$. $\mathbf{V_{HC}}$ is the helix spiral vector at the beginning of the C-terminal helix, which was identified as the vector from the first Cα atom (red) in the C-terminal helix to the Cα atom immediately after the first Cα atom (yellow). $\phi_{CL}$ was identified as the dihedral angle between the plane defined with the $\mathbf{V_C}$ and $\mathbf{V_{HC}}$ vectors and that with the $\mathbf{V_L}$ and $\mathbf{V_{HC}}$ vectors.

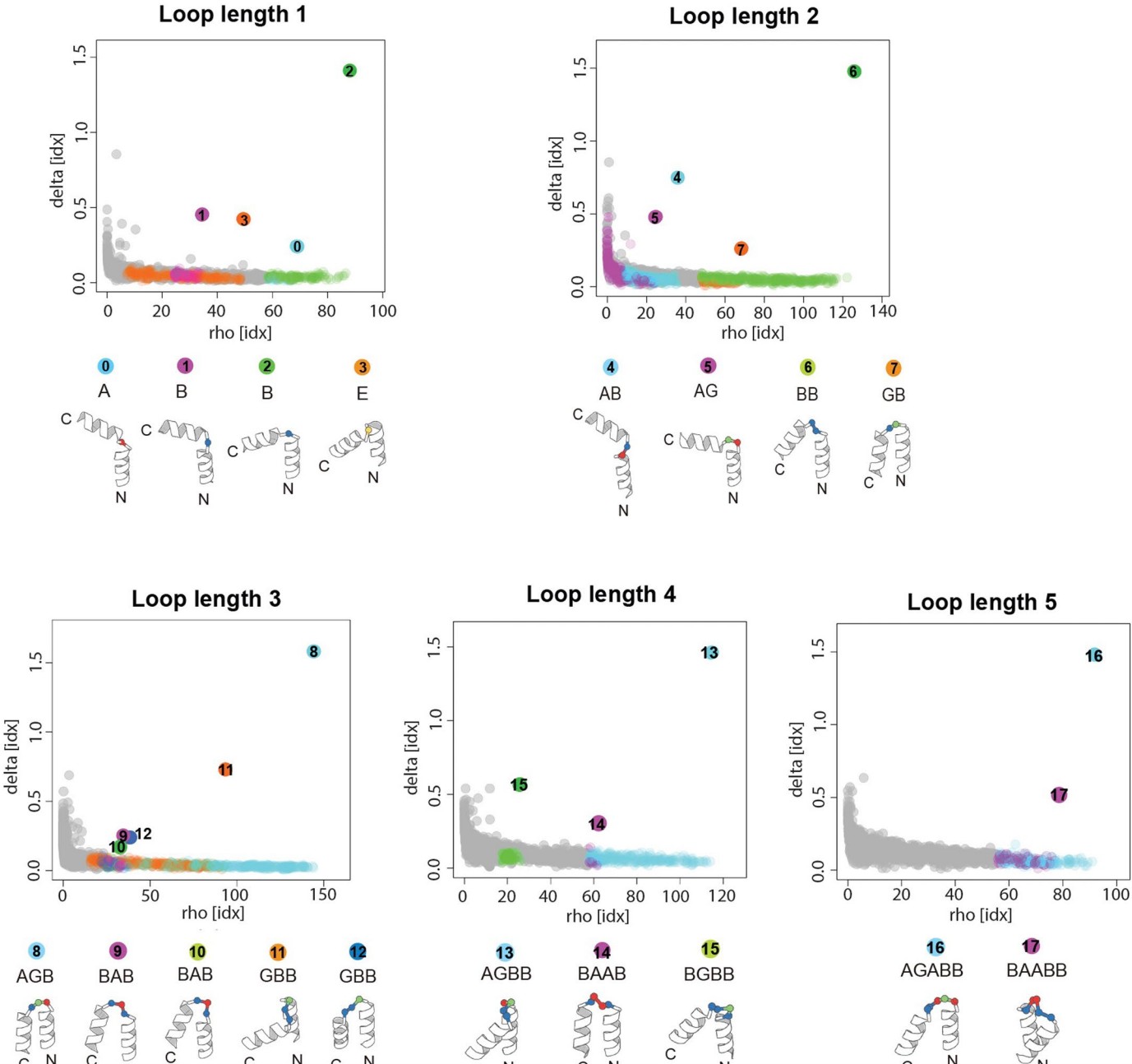

**Extended Data Fig. 3 | Statistical analysis of HLH motifs of naturally occurring proteins.** HLH structures collected from naturally occurring protein structures were clustered for each loop length from one to five based on the pairwise Euclidean distance between the five-dimensional vectors of the features shown in Extended Data Fig. 2, using the density clustering algorithm[30]. For each loop length, decision graphs to determine density peaks of clusters are shown, in which rho represents the local density of a point in the five-dimensional feature vector space and delta represents the minimum distance between a point to any other point with higher density; for the point with highest density, delta is calculated as the maximum distance to any other points.

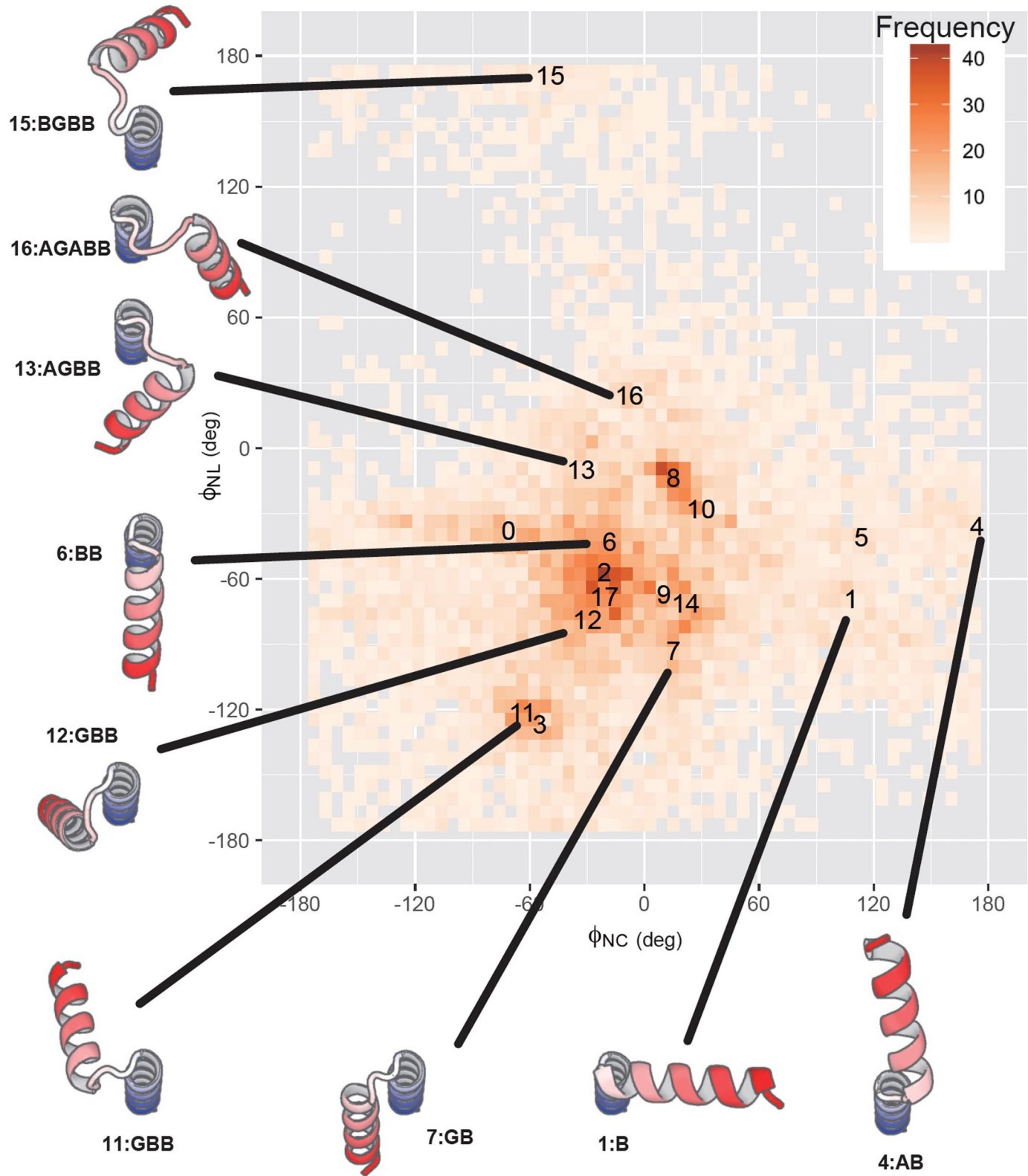

**Extended Data Fig. 4 | Mapping of 18 representative HLH motifs on the $\phi_{NC}$-$\phi_{NL}$ plane.** The loop numbers with their ABEGO torsion patterns correspond to the 18 representative HLH motifs shown in Extended Data Fig. 3.

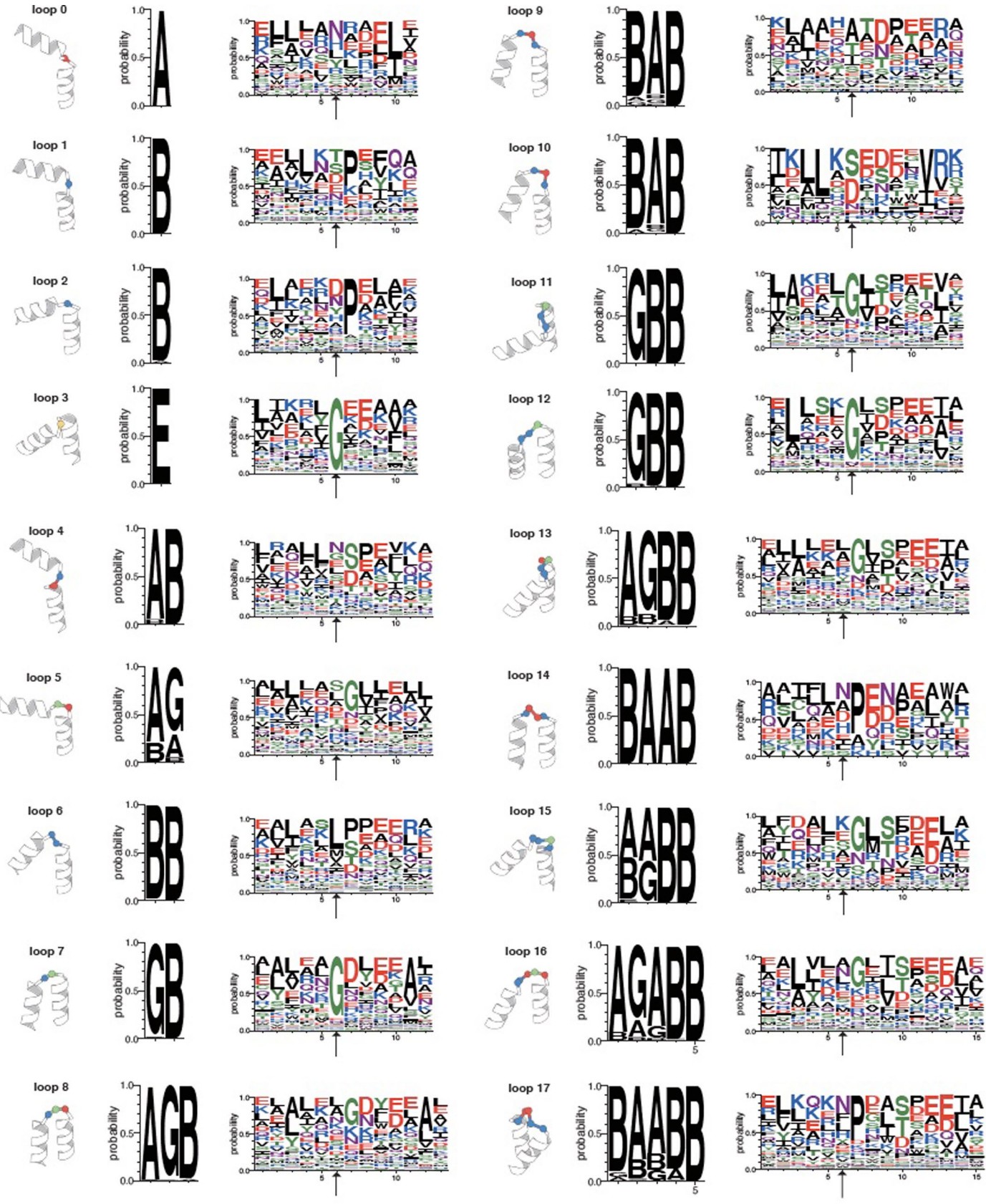

**Extended Data Fig. 5 | ABEGO-based loop geometries and amino acid sequence preferences of the cluster that each HLH motif belongs to.** (Left) The 18 representative HLH motifs are shown as in Extended Data Fig. 3. (Middle) ABEGO torsion patterns of the loop. This result suggests that the relative arrangements of adjacent helices strongly limit the torsion patterns of the connecting loop. (Right) Amino acid sequence preferences of each HLH motif. The first residue of the loop is indicated by an arrow.

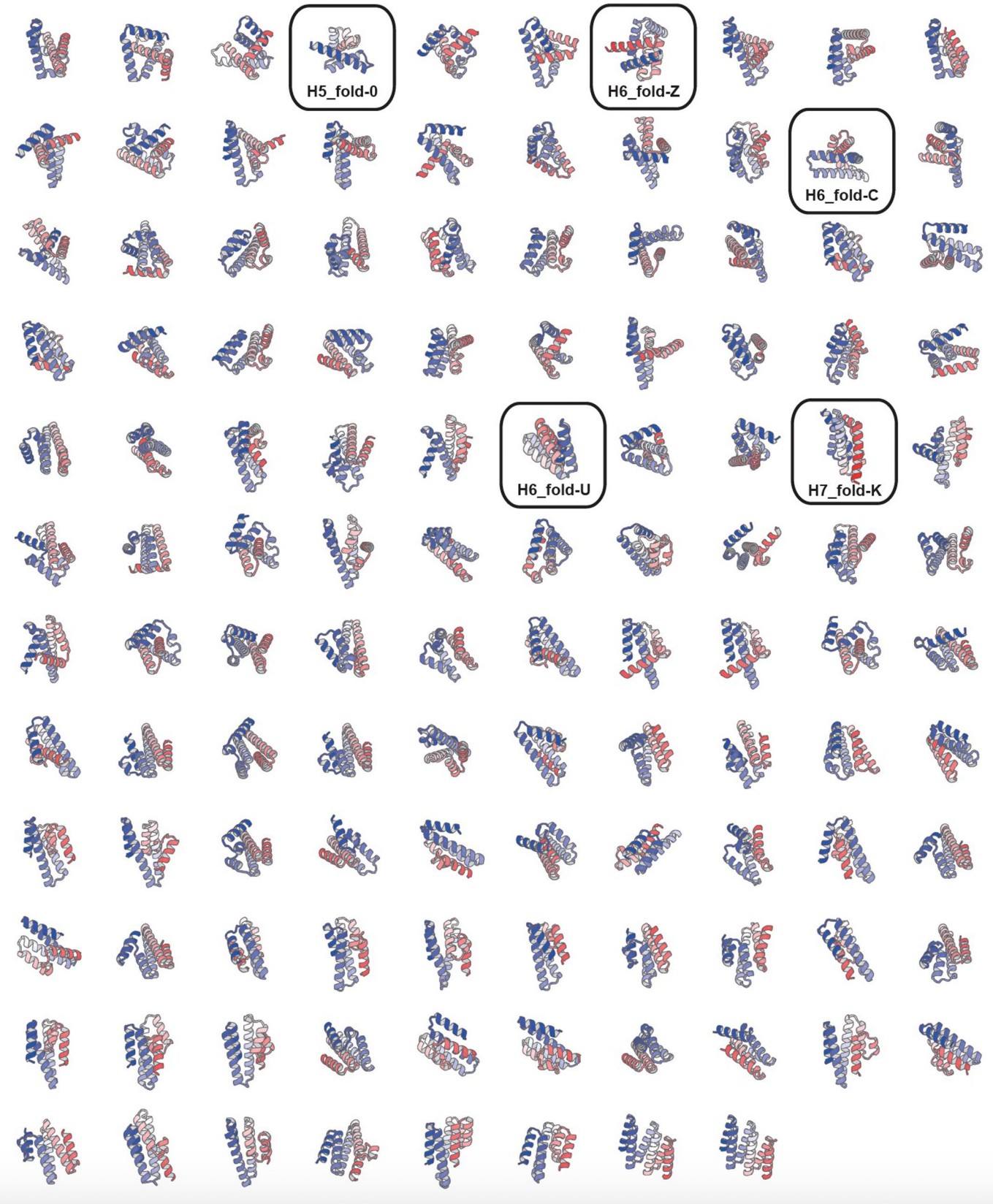

**Extended Data Fig. 6 | Examples of compact structures obtained from the enumeration of six-helix structures.** The structures are sorted by their HO values from top left to bottom right. The top left structure shows the smallest HO value and has irregularly packed α-helices, whereas the bottom right one shows the highest HO value and has parallelly aligned α-helices. The five designed topologies are shown together by enclosed squares.

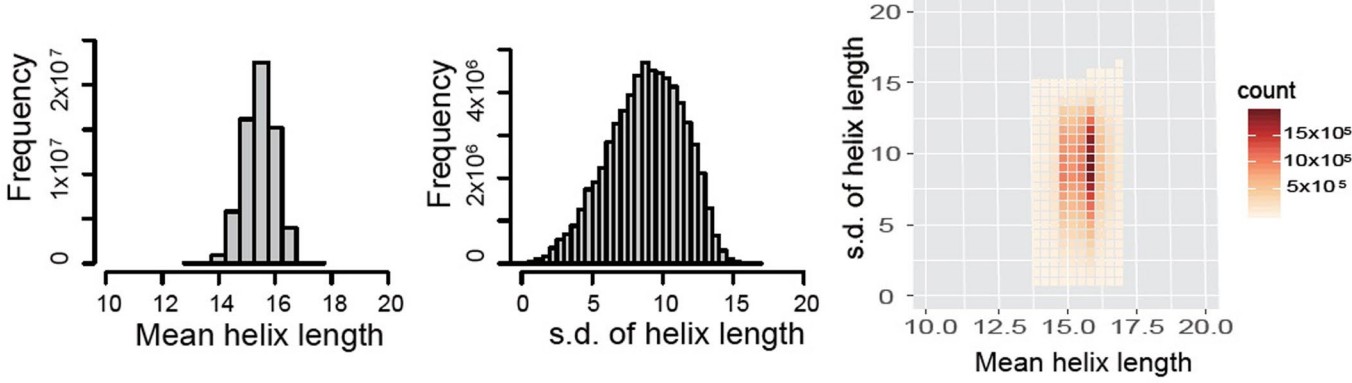

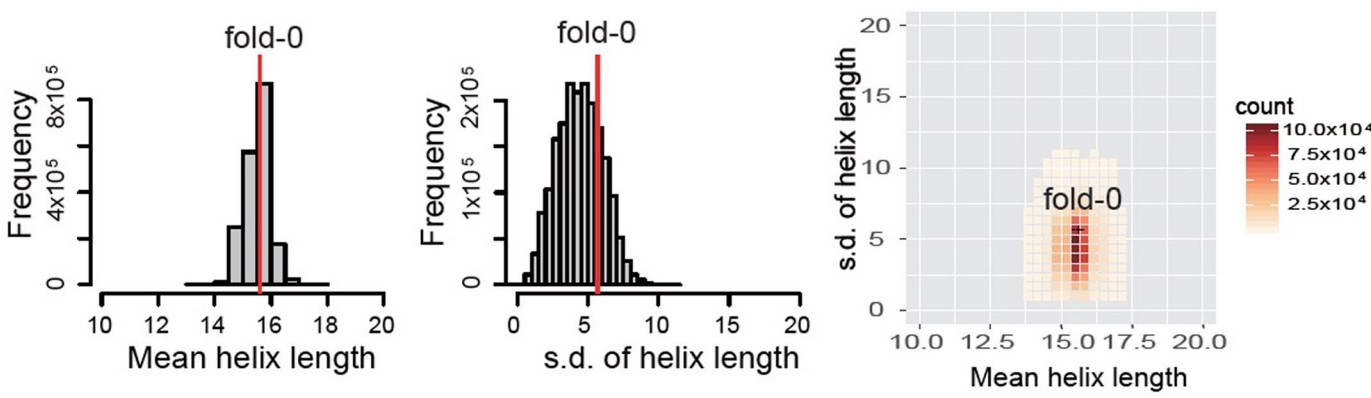

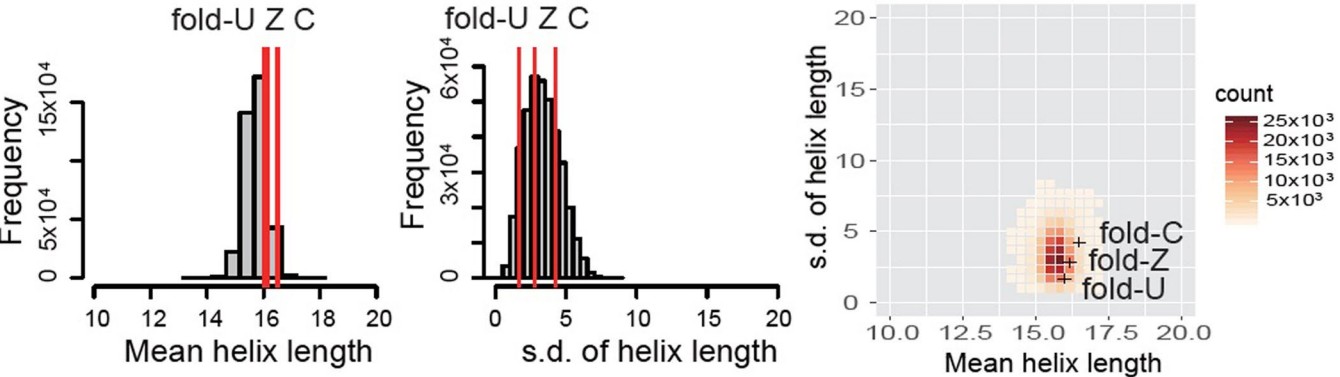

**Extended Data Fig. 7 | Distribution of the helix lengths for the generated backbone structures with four-helix and 70 residues, five-helix and 90 residues, and six-helix and 110 residues.** Left: The distribution of mean helix lengths for the generated backbone structures. For each backbone structure, the mean helix length was calculated by averaging lengths of the helices in the structure. Middle: The distribution of standard deviation of helix lengths for the generated backbone structures. For each backbone structure, the standard deviation of lengths of the helices in the structure was calculated. Right: The two-dimensional distribution of the mean and standard deviation of helix lengths.

The width of the distribution of the standard deviation was in the order of the four-helix, five-helix, and six-helix structures. This is because the four-helix structures were not subject to the Rg constraint, and the five- and six-helix structures were the ones with $Rg < 14$ Å. Since the same threshold value for the Rg constraint was used, the distribution width for the five-helix structures is slightly wider than that of the six-helix structures. The helix lengths of the designs chosen for experimental characterization correspond to the vicinity of the peaks of the distributions.

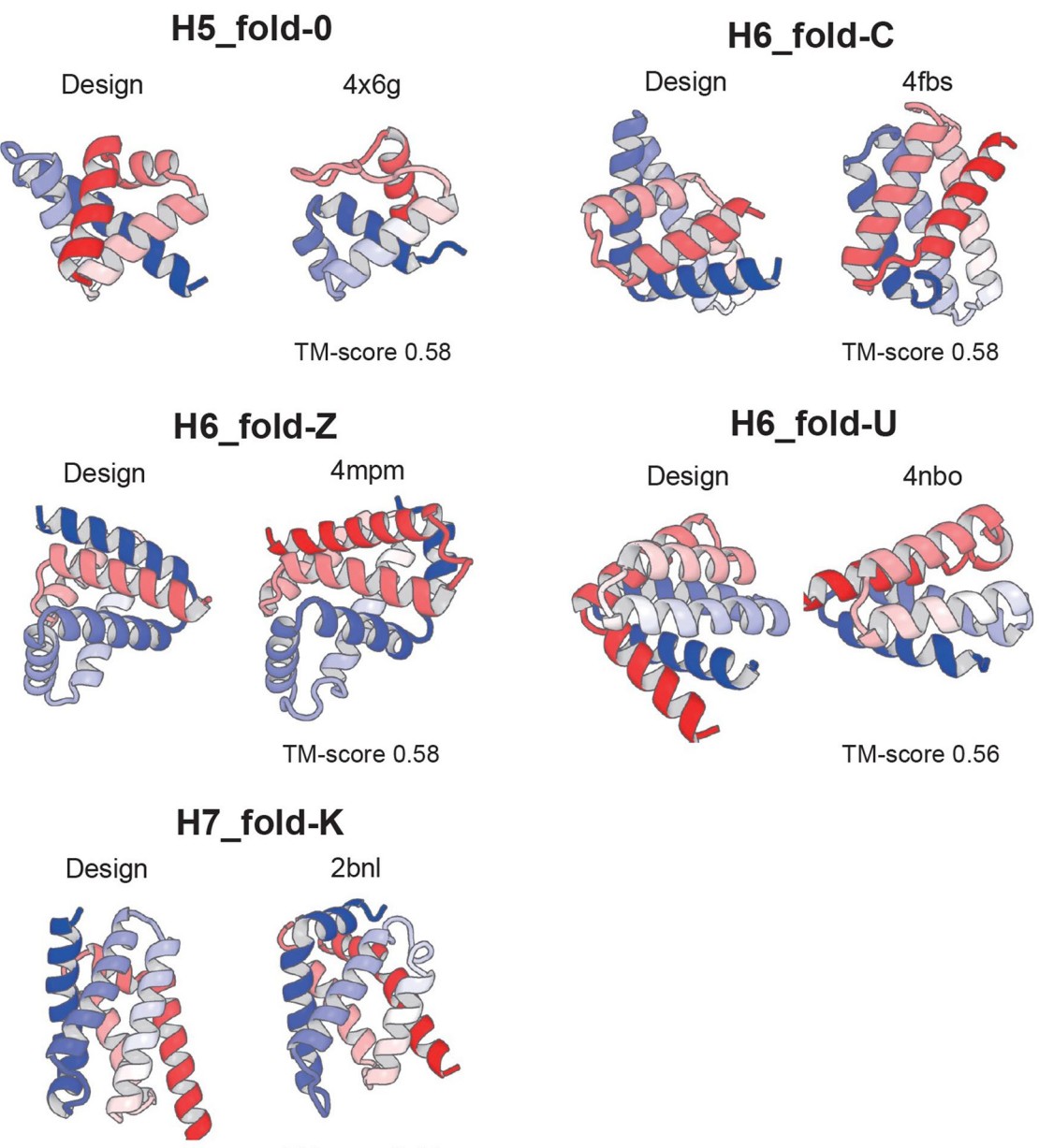

**Extended Data Fig. 8 | Comparison of designed structures and the most similar naturally occurring proteins.** The designed structures (left) and the most similar ones (right) with pdb ids and TM-score values.

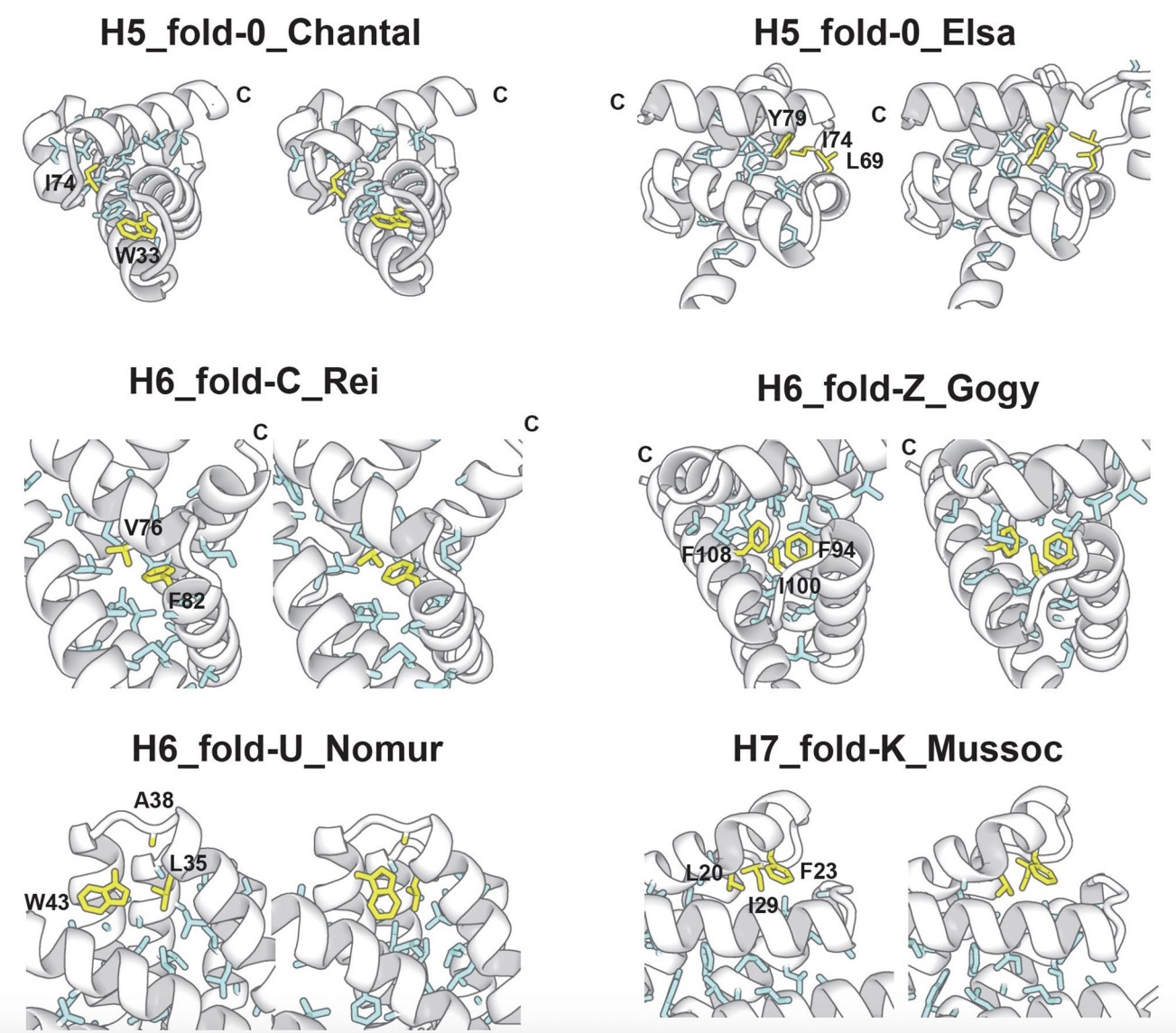

**Extended Data Fig. 9 | Comparison of computational models (left) with experimentally determined structures (right).** Hydrophobic core residues are shown in stick. Bulky hydrophobic side chains from loops and the neighboring α-helices, which spiked the core and pinned the loops to the target conformations, are shown in yellow.

**Extended Data Table 1 | Summary of experimental results for designed proteins**

| | #designs tested | Expressed† | Soluble† | α-protein CD spectrum (20 °C) | Monomeric‡ | Well resolved HSQC* | Success (rate %) |
|---|---|---|---|---|---|---|---|
| **H5_fold-0** | 10 | 10 | 9 | 9 | 7 | 6 | 6 (60) |
| **H6_fold-C** | 7 | 7 | 6 | 6 | 5 | 3 | 3 (43) |
| **H6_fold-Z** | 7 | 7 | 7 | 7 | 4 | 4 | 4 (57) |
| **H6_fold-U** | 8 | 8 | 5 | 5 | 5 | 4 | 4 (50) |
| **H7_fold-K** | 8 | 8 | 7 | 7 | 6 | 6 | 6 (75) |

The second column shows the number of designs experimentally tested for the fold in the leftmost column. The subsequent columns give the number of designs that satisfy each experimental characterization, which was performed sequentially from the left to the right. The successful designs are defined as those that satisfy all criteria and are expected to fold into correct fold. The details of the results are shown in Supplementary Tables 2-6. †Expression and solubility were assessed by SDS-PAGE and mass spectrometry. ‡SEC-MALS was used to determine oligomerization state. The number of designs in which the main peak of the absorbance at 280 nm corresponds to the monomeric state was counted. *1H-15N HSQC spectra were collected.

# Reporting Summary

## Statistics

For all statistical analyses, confirm that the following items are present in the figure legend, table legend, main text, or Methods section.

| n/a | Confirmed | |
|---|---|---|
| ☐ | ☒ | The exact sample size (*n*) for each experimental group/condition, given as a discrete number and unit of measurement |
| ☐ | ☒ | A statement on whether measurements were taken from distinct samples or whether the same sample was measured repeatedly |
| ☒ | ☐ | The statistical test(s) used AND whether they are one- or two-sided<br>*Only common tests should be described solely by name; describe more complex techniques in the Methods section.* |
| ☒ | ☐ | A description of all covariates tested |
| ☒ | ☐ | A description of any assumptions or corrections, such as tests of normality and adjustment for multiple comparisons |
| ☐ | ☒ | A full description of the statistical parameters including central tendency (e.g. means) or other basic estimates (e.g. regression coefficient) AND variation (e.g. standard deviation) or associated estimates of uncertainty (e.g. confidence intervals) |
| ☒ | ☐ | For null hypothesis testing, the test statistic (e.g. *F*, *t*, *r*) with confidence intervals, effect sizes, degrees of freedom and *P* value noted<br>*Give P values as exact values whenever suitable.* |
| ☒ | ☐ | For Bayesian analysis, information on the choice of priors and Markov chain Monte Carlo settings |
| ☒ | ☐ | For hierarchical and complex designs, identification of the appropriate level for tests and full reporting of outcomes |
| ☒ | ☐ | Estimates of effect sizes (e.g. Cohen's *d*, Pearson's *r*), indicating how they were calculated |

*Our web collection on statistics for biologists contains articles on many of the points above.*

## Software and code

Policy information about availability of computer code

| | |
|---|---|
| Data collection | The code for building helical backbone structures has been implemented into Rosetta at https://github.com/RosettaCommons/main/tree/koga/all-alpha_design. The demo for building helical structures will be available at https://github.com/kogalab21/all-alpha_design. Rosetta software suite 3 was used for protein design and folding calculations. JASCO SpectraManager software v2 was used for CD. |
| Data analysis | Analyses on helical backbone structures were carried out with Rosetta. The code for analyzing helical structures has been implemented into Rosetta at https://github.com/RosettaCommons/main/tree/koga/all-alpha_design.<br>Thermal denaturation data by CD were fit using nls function in R programming 3.3.1.<br>SEC-MALS data were analyzed by the ASTRA software 6.1.2.<br>HSQC data were analyzed by the Delta 5.0.4 NMR softwares.<br>All NMR structure analyses were done as described in the methods section with the following programs: MagRO-NMRViewJ (updated version of Kujira),Filt_Robot, TALOS+ 2017, FLYA 3.98.5, CYANA 3.98.5, Amber 12, and PALES 2.1.<br>The X-ray structure analysis was done as described in the methods section with the following programs: XDS VERSION Jan 26, 2018 BUILT=20180126, CCP4  7.0.052, Coot 0.8.1, Phenix Refine 1.12, and RAMPAGE (CCP4: 7.0.053). |

For manuscripts utilizing custom algorithms or software that are central to the research but not yet described in published literature, software must be made available to editors and reviewers. We strongly encourage code deposition in a community repository (e.g. GitHub). See the Nature Portfolio guidelines for submitting code & software for further information.

## Data

Policy information about availability of data

All manuscripts must include a data availability statement. This statement should provide the following information, where applicable:

- Accession codes, unique identifiers, or web links for publicly available datasets
- A description of any restrictions on data availability
- For clinical datasets or third party data, please ensure that the statement adheres to our policy

The solution NMR structures have been deposited in the wwPDB as PDB 7BQM (H5_fold-0_Chantal), 7BQN (H6_fold-C_Rei), 7BQQ (H6_fold-Z_Gogy), 7BQS (H6_fold-U_Nomur), and 7BQR (H7_fold-K_Mussoc). The NMR data were deposited in the BMRB under the accession numbers 36335 (H5_fold-0_Chantal), 36336 (H6_fold-C_Rei), 36337 (H6_fold-Z_Gogy), 36339 (H6_fold-U_Nomur), and 36338 (H7_fold-K_Mussoc). The crystal structure of H5_fold-0_Elsa has been deposited in the wwPDB as 7DNS. The computational design models are presented as Supplementary Data 1. The generated compact and steric-clash-free five-helix (1,899,355) and six-helix (380,869) structures are available at https://github.com/kogalab21/all-alpha_design. The plasmids encoding the designed sequences are available through Addgene under the accession numbers 201825 (H5_fold-0_Elsa), 201826 (H5_fold-0_Chantal), 201827 (H6_fold-C_Rei), 201828 (H6_fold-Z_Gogy), 201829 (H6_fold-U_Nomur), and 201830 (H7_fold-K_Mussoc). Source Data are available at https://github.com/kogalab21/all-alpha_design.

# Field-specific reporting

Please select the one below that is the best fit for your research. If you are not sure, read the appropriate sections before making your selection.

☒ Life sciences  ☐ Behavioural & social sciences  ☐ Ecological, evolutionary & environmental sciences

For a reference copy of the document with all sections, see nature.com/documents/nr-reporting-summary-flat.pdf

# Life sciences study design

All studies must disclose on these points even when the disclosure is negative.

| | |
|---|---|
| Sample size | Computational designs that passed selection criteria were experimentally tested. Based on the previously reported success rate of de novo designed proteins (N. Koga et al. Nature, 2012; Y.-R. Lin et al., PNAS, 2015), we estimated the number of designs we should test in order to be successful. |
| Data exclusions | No data were excluded |
| Replication | The representative designs for NMR structure determination were purified, verified by SDS-PAGE, mass spectrometry, and HSQC measurements twice independently. All attempts at replication were successful. |
| Randomization | Randomization is not relevant to our study. This is an observational study, which does not involve evaluation of conditional effects. |
| Blinding | Blinding is not relevant to our study. Keeping track of the identity of each designed protein was necessary for characterizing biophysical properties and solving the structure. |

# Reporting for specific materials, systems and methods

We require information from authors about some types of materials, experimental systems and methods used in many studies. Here, indicate whether each material, system or method listed is relevant to your study. If you are not sure if a list item applies to your research, read the appropriate section before selecting a response.

## Materials & experimental systems

| n/a | Involved in the study |
|---|---|
| ☒ ☐ | Antibodies |
| ☒ ☐ | Eukaryotic cell lines |
| ☒ ☐ | Palaeontology and archaeology |
| ☒ ☐ | Animals and other organisms |
| ☒ ☐ | Human research participants |
| ☒ ☐ | Clinical data |
| ☒ ☐ | Dual use research of concern |

## Methods

| n/a | Involved in the study |
|---|---|
| ☒ ☐ | ChIP-seq |
| ☒ ☐ | Flow cytometry |
| ☒ ☐ | MRI-based neuroimaging |

