## [Peer Review File · Nature Structural & Molecular Biology]

Peer Review Information

Manuscript Title: Design of complicated all- α protein structures

Corresponding author name(s): Nobuyasu Koga

Reviewer Comments & Decisions:

Decision Letter, initial version:
--

Message: 7th Jan 2022

Dear Dr. Koga,

Thank you again for submitting your manuscript "Design of complicated all- α protein structures". I apologize for the delay in responding, which resulted from the difficulty in obtaining suitable referee reports. Nevertheless, we now have comments (below) from the 2 reviewers who evaluated your paper. In light of those reports, we remain interested in your study and would like to see your response to the comments of the referees, in the form of a revised manuscript.

You will see that the referees asked for additional explanations about the criteria used to assess the properties of the newly designed folds. Please be sure to address/respond to all concerns of the referees in full in a point-by-point response and highlight all changes in the revised manuscript text file. If you have comments that are intended for editors only, please include those in a separate cover letter.

We expect to see your revised manuscript within 6 weeks. If you cannot send it within this time, please contact us to discuss an extension; we would still consider your revision, provided that no similar work has been accepted for publication at NSMB or published elsewhere.

Reporting Summary:

Please note that all key data shown in the main figures as cropped gels or blots should be presented in uncropped form, with molecular weight markers. These data can be aggregated into a single supplementary figure item. While these data can be displayed in a relatively informal style, they must refer back to the relevant figures. These data should be submitted with the final revision, as source data, prior to acceptance, but you may want to start putting it together at this point.

Data availability: this journal strongly supports public availability of data. All data used in accepted papers should be available via a public data repository, or alternatively, as Supplementary Information. If data can only be shared on request, please explain why in your Data Availability Statement, and also in the correspondence with your editor. Please note that for some data types, deposition in a public repository is mandatory - more

information on our data deposition policies and available repositories can be found below:
<https://www.nature.com/nature-research/editorial-policies/reporting-standards#availability-of-data>

[redacted]

Sincerely,
Sara

Sara Osman, Ph.D.
Associate Editor
Nature Structural & Molecular Biology

Referee expertise:

Referee #1: Protein folding, NMR, biophysical methods, MD

Referee #2: De novo protein design

Reviewers' Comments:

Reviewer #1:

Remarks to the Author:

This manuscript reports the de novo design and experimental characterization of proteins consisting of 5 or 6 alpha-helices connected up 1-5 residue loops, with topologies ranging from bundle-like to more complex. Several of these complex topologies differ from most previous de novo helical protein designs, which have predominantly parallel arrangements of helices, and so approach high complexity analogous to that observed in natural proteins, such as globins. This complexity arises from building a wide range of topologies out of common helix-loop-helix motifs found in natural proteins. This new design strategy is clever and has broader implications, valuable progress. The strategy is demonstrated for 5 new folds, with the designed proteins having very high apparent melting temperatures and experimental structures generally close ($<2\text{\AA}$ and up to $\sim 3\text{\AA}$ rmsd) to the design structures. The work is generally clearly presented and appropriated credits previous work. Thus, overall, this manuscript represents an important advance. The points below should be addressed prior to publication.

1. An important point to consider in additional detail is the criterion (or criteria) that may best be used as a measure of protein compactness. The radius of gyration is used here, but this criterion is just 1 global value, which has potential to overlook rather poor packing of specific helices, e.g. as in H6_fold-U (though in Extended Data Fig 7 H6_fold-C has multiple long helices that seem to jut out of the structure, see also next point) . Please expand explanation and justification for using Rg (e.g. how values may compare with other proteins, of varying sizes etc). It may be valuable to include additional measures of compactness or packing in design?
2. Additional significant points that warrant further explanation are how was varying the length of the canonical helices implemented and to what extent were different helix lengths explored during computational design and in the designs chosen for experimental characterization.
3. The preceding points relate to the more general question of just how cooperative is the un/folding of the different designs? The strongly sloping baselines and small amplitude of the sigmoidal unfolding region in the CD data for H6_fold-U might indicate a gradual loss of helical structure (for example in peripheral helices) in the native baseline, followed by cooperative unfolding of a core of several helices, and then further loss of significant residual helical structure in the denatured state. To facilitate comparing data, I recommend to use the same scales for the plots of CD data. Along with further consideration of CD data, the changes in heat capacity of unfolding (was thermal unfolding reversible?) and NMR data (see below) may inform on designed vs experimentally achieved cooperativity.
4. Line 130: please comment on what proportion of designs may show funnel-shaped energy landscapes
5. Line 143: well-dispersed sharp peaks for 22 designs. It is not so clear that peaks are well-dispersed for all the designs. While helical proteins tend in general to have relatively low dispersion, some of the designs appear to have fewer peaks than expected, perhaps due to overlapped peaks in random coil regions or due to peak broadening arising from

conformational exchange or protein self-association. Expanding on these points is very relevant to better define and understand just how well folded and cooperative the designs are. Information on the number and identities of assigned residues should be added, e.g. in Supp Table 7 or other easily accessible place)

6. Line 175 (minor point): specify which 3 designs

7. Line 181: for clarity, incorporate and indicate the 5 experimentally characterized designs into Extended Data Fig. 6.

8. Lines 183-188: some additional supporting references should be added.

Reviewer #2:

Remarks to the Author:

This paper by Sakuma et al describes the de novo design of complex all-alpha helical folds using a fragment assembly strategy with parts derived from analysis of helix-turn-helix structures in the PDB. They report a highly impressive success rate with regards to NMR analysis (22/40 designs exhibiting excellent peak dispersion) and the high resolution structures (NMR and 1 X-ray) for a representative of the selected folds. The robustness of the methodology is apparent in the experimental results and reported success rates, and its publication will aid the expansion of this approach to future functional proteins and enzymes. Beyond the minor issues listed below, my only criticism lies in the lack of discussion around the HLH sequences, the sequence consensus (or lack of it) in extended figure 5, and I would ideally like to know more about the energy landscapes of these helix-turn-helix sequences/structures, and about the physical basis of the sequence-structure relationships. This is likely beyond what can be discussed within a paper of this length, but perhaps could be commented on in the manuscript.

It is therefore my opinion that this work represents a significant advance in de novo protein design, and I recommend it for publication in NSMB following the correction of minor errors/typos.

Line 43 - this method...will enable us...

Line 52 - 'and so are many naturally occurring proteins' doesn't make sense in the context of the sentence

Line 57 - the word 'parallelly' doesn't look right to me - suggest rewording sentence - '...consisting of α -helices in almost parallel alignment...'

Line 61 - 'All α -proteins...' could do with a reference

Line 68 - the introduction ends abruptly, and usually a summary of the work to be undertaken is included at this stage - is this a constraint of the format?

Line 72 - I don't agree that a major obstacle in designing all- α structures is one of imagination - we have plenty of examples of such structures in nature (i.e. the globin fold!) and this statement is rather ambiguous. Do the Authors mean that they cannot a priori determine the feasibility of helical arrangements at this stage? If so, then they should state that. If not, they need to clarify what they mean.

Line 81 - Sentence would be improved by beginning 'Therefore, the question is whether...'

Line 120 - 'None of these backbone structures are similar...'

Line 154 - 'were found to be coherently packed to constitute...'

Line 187 - 'which enables to locally change the conformations,...' needs to be reworded

Line 676 - Figure legend should indicate that the CD spectra are recorded under high pressure

Line 784 - Typo - change 'balky' to 'bulky'

Author Rebuttal to Initial comments

Dear Dr. Sara Osman,

Thank you very much for reviewing our manuscript (NSMB-A45429-T) entitled “Design of complicated all- α protein structures” by Koya Sakuma, Naohiro Kobayashi, Toshihiko Sugiki, Toshio Nagashima, Toshimichi Fujiwara, Kano Suzuki, Naoya Kobayashi, Takeshi Murata, Takahiro Kosugi, Rie Koga, and Nobuyasu Koga. We sincerely appreciate all reviewer’s constructive and insightful comments and suggestions. We have revised our manuscript according to them as described in the following pages.

Nobuyasu Koga Associate Professor,
National Institutes of Natural Sciences,
Exploratory Research Center on Life and Living Systems (ExCELLS)
38 Nishigo-Naka, Myodaiji, Okazaki, 444-8585, Japan.
Phone: +81-564-55-7379 E-Mail: nkoga@ims.ac.jp

To Reviewer #1:

1-1) An important point to consider in additional detail is the criterion (or criteria) that may best be used as a measure of protein compactness. The radius of gyration is used here, but this criterion is just 1 global value, which has potential to overlook rather poor packing of specific helices, e.g. as in H6_fold-U (though in Extended Data Fig 7 H6_fold-C has multiple long helices that seem to jut out of the structure, see also next point). Please expand explanation and justification for using Rg (e.g. how values may compare with other proteins, of varying sizes etc). It may be valuable to include additional measures of compactness or packing in design?

Thank you very much for the comments and suggestions. We agree with you that the radius of gyration is just one of the parameters to describe “compactness” of protein structures. As you pointed out, packing is also an important parameter to describe the compactness. However, the tight core packings, such as those observed in naturally occurring proteins, are created through the sidechain design procedure: the iteration of fixed-backbone sequence (sidechains) optimization and fixed-sequence structure optimization; we are not able to evaluate the packing in advance for backbone structures generated by our developed strategy. Therefore, we used the radius of gyration (Rg) as a simple parameter for selecting backbone structures. The tight core sidechain packing of designed proteins including the one for H6_fold-U was confirmed using RosettaHoles after the design procedure (different from this discussion, a poorly packed terminal helix was observed in the NMR structures for H6_fold-U_Nomor, which was described in the response 1-3-2).

To provide a justification for the threshold of Rg value we used in this work, we investigated the Rg

distribution for naturally occurring proteins and added Extended Data Fig. 18. The distribution shows the peak at $\sim 14 \text{ \AA}$, which corresponds to the threshold we used for the backbone selection; the R_g values of the five designs correspond to the vicinity of the peak. Accordingly, we added the explanation for using the threshold into the section “Building backbone structures” in the Methods (Lines: 367-368), and added the R_g values for the design target backbone structures shown in Fig. 4.

Despite the above discussion, one of the open questions is whether all or how much of the backbone structures generated by our strategy can have tight core packing. In this study, we tested only five backbone structures; the packability for the other backbone structures has not been clarified, which should be addressed in next works. We added this question into the second paragraph in the Discussion (Lines: 211-215).

1-2) Additional significant points that warrant further explanation are how was varying the length of the canonical helices implemented and to what extent were different helix lengths explored during computational design and in the designs chosen for experimental characterization.

Thank you for the comments. Helix lengths were exhaustively explored from 5 to 35 residues to identify compact and steric-clash-free backbone structures. We investigated helix length distributions of the generated backbone structures for 4-, 5-, and 6- helix structures, and found that the helix lengths are broadly distributed from short to long. The width of the helix length distribution was in the order of the 4-helix, 5-helix, and 6-helix structures. This is because the 4-

helix structures were not subject to the R_g constraint, and the 5- and 6- helix structures were the ones with $R_g < 14 \text{ \AA}$. Since the same threshold value for the R_g constraint was used, the distribution width for the 5-helix structures is slightly wider than that of the 6-helix structures. The helix lengths of the designs chosen for experimental characterization correspond to the vicinity of the peaks of the distributions. We added one sentence about the distribution of helix lengths (Lines: 117-118) and Extended Data Fig. 7.

1-3) The preceding points relate to the more general question of just how cooperative is the un/folding of the different designs? The strongly sloping baselines and small amplitude of the sigmoidal unfolding region in the CD data for H6_fold-U might indicate a gradual loss of helical structure (for example in peripheral helices) in the native baseline, followed by cooperative unfolding of a core of several helices, and then further loss of significant residual helical structure in the denatured state.

Thank you for the comments. We described this in the response, 1-3-2.

1-3-1) To facilitate comparing data, I recommend to use the same scales for the plots of CD data.

Following your suggestion, we replotted the CD data in Figure 5.

1-3-2) Along with further consideration of CD data, the changes in heat capacity of unfolding (was thermal unfolding reversible?) and NMR data (see below) may inform on designed vs experimentally achieved cooperativity.

Thank you for the suggestions. We studied the number of local and non-local distance constraints for each residue obtained for the NMR structure determination. For the C-terminal helix of H6_fold-U_Nomur, the number of local distance constraints indicated the helix formation.

	T_m (°C)	ΔH (kcal/mol)	ΔC_p (kcal/mol/K)
H5_fold-0_Chantal	117.8 ± 0.2	105.2 ± 5.2	2.00 ± 0.13
H5_fold-0_Elsa	105.2 ± 0.2	91.2 ± 5.5	1.94 ± 0.20
H6_fold-C_Rei	105.0 ± 0.2	97.9 ± 5.4	1.97 ± 0.79
H6_fold-Z_Gogy	122.4 ± 0.2	105.6 ± 5.1	0.06 ± 2.00
H6_fold-U_Nomur	116.3 ± 0.3	98.9 ± 8.1	1.73 ± 1.02

However, the number of non-local distance constraints for that helix was less than that of the other helices of the five designed proteins. This result indicates that the C-terminal helix of H6_fold-U_Nomur is loosely packed despite the helix formation. We added the Extended Data Figs. 15-16.

Next, we considered the CD data. Since thermal unfolding up to immediately after transition was almost reversible for all designs except H7_fold-K_Mussoc, we carried out the fitting of their thermal unfolding curves to the Gibbs-Helmholtz equation using the approximation by a two-state unfolding model. The results are shown in the following table.

However, the ΔC_p values for Rei, Gogy, and Nomur were not obtained with accuracy. Moreover, the ΔH values were almost the same for all designs. This may be because the temperature range of the transition for H6_fold-U_Nomur was similar to those of the other designs, despite a gradual loss of helical structures along the native baseline and a further loss of residual helical structures along the denatured baseline. Therefore, we then investigated the cooperativity of the designs using chemical denaturation; the results were added into Fig. 5d. We found that the m -value of H6_fold-U_Nomur,

representing the cooperativity, was lower than those of the other designs (Note that the H5_fold-0_Elsa and Chantal, which are smaller in size than the other designs, show lower m -values. This is because m -values also depend on protein size, with larger proteins having larger m -values), which would be due to the loose packing of the C-terminal helix of H6_fold-U_Nomur.

We added the above discussion in the section “Experimental characterization of designed proteins” in the Results (Lines: 174-182, 185-186).

1-4) Line 130: please comment on what proportion of designs may show funnel-shaped energy landscapes

Following your suggestion, we added the sentence on what proportion of designs showed funnel-shaped energy landscapes (Lines: 140-143).

1-5) Line 143: well-dispersed sharp peaks for 22 designs. It is not so clear that peaks are well-dispersed for all the designs. While helical proteins tend in general to have relatively low dispersion, some of the designs appear to have fewer peaks than expected, perhaps due to overlapped peaks in random coil regions or due to peak broadening arising from conformational exchange or protein self-association. Expanding on these points is very relevant to better define and understand just how well folded and cooperative the designs are. Information on the number and identities of assigned residues should be added, e.g. in Supp Table 7 or other easily accessible place)

Following your suggestion, we added Extended Data Fig. 10 for HSQC spectra for 23 designs (we corrected the number of designs from 22 to 23) and Supplementary Table 8 for the number and identities of assigned atoms in the NMR structure determination. We found that the HSQC spectra for the 23 designs showed well-dispersed sharp peaks typically observed for helical proteins, and almost all atoms were assigned in the NMR structure determination. Therefore, together with the results of CD and SEC-MALS measurements, we think that the 23 designs mostly form stable and monomeric tertiary structures, although some partial structures may be fluctuated, as observed in the C-terminal helix of H6_fold-U_Nomur.

1-6) Line 175 (minor point): specify which 3 designs

Thank you for pointing this out. We specified them (Line: 205).

1-7) Line 181: for clarity, incorporate and indicate the 5 experimentally characterized designs into Extended Data Fig. 6.

Thank you for pointing this out. We added the 5 designs into Extended Data Fig. 6.

1-8) Lines 183-188: some additional supporting references should be added.

Thank you for the suggestion. However, we were not able to find appropriate references to support the hypothesis, and realized that this should be clarified by future de novo design studies. To convey the message, we modified the discussion (Lines: 217-219).

To Reviewer #2:

This paper by Sakuma et al describes the de novo design of complex all-alpha helical folds using a fragment assembly strategy with parts derived from analysis of helix-turn-helix structures in the PDB.

2-1) Beyond the minor issues listed below, my only criticism lies in the lack of discussion around the HLH sequences, the sequence consensus (or lack of it) in extended figure 5, and I would ideally like to know more about the energy landscapes of these helix-turn-helix sequences/structures, and about the physical basis of the sequence-structure relationships. This is likely beyond what can be discussed within a paper of this length, but perhaps could be commented on in the manuscript.

Thank you for the comments. We added descriptions for the sequence-structure relations for the HLH motifs (Lines: 100-102). Moreover, we investigated the sequence-structure relations for the HLH motifs in terms of hydrophobic or helix-capping residues by introducing Ala mutations at those residue positions and exploring energy landscapes for the mutants. We added the descriptions (Lines: 172-173, 184-185) and Extended Data Fig. 14.

2-2) It is therefore my opinion that this work represents a significant advance in de novo protein design, and I recommend it for publication in NSMB following the correction of minor errors/typos.

2-2-1) Line 43 - this method...will enable us...

We modified it (Line: 45).

2-2-2) Line 52 - 'and so are many naturally occurring proteins' doesn't make sense in the context of the sentence

We reworded it to 'and many naturally occurring proteins as well' (Lines: 55-56).

2-2-3) Line 57 - the word 'parallely' doesn't look right to me - suggest rewording sentence -
'...consisting of a-helices in almost parallel alignment...'

Following your suggestion, we reworded it (Line:59). Thank you.

2-2-4) Line 61 - 'All a-proteins...' could do with a reference

Thank you for the comments. However, we were not able to find appropriate references to support the hypothesis. Therefore, we modified the introduction and described the sentences as an assumption (Lines: 62-66).

2-2-5) Line 68 - the introduction ends abruptly, and usually a summary of the work to be undertaken is included at this stage - is this a constraint of the format?

Following your suggestion, we added a summary as the last paragraph in the Introduction (Lines: 71-76).

2-2-6) Line 72 - I don't agree that a major obstacle in designing all- α structures is one of imagination - we have plenty of examples of such structures in nature (i.e. the globin fold!) and this statement is rather ambiguous. Do the Authors mean that they cannot a priori determine the feasibility of helical arrangements at this stage? If so, then they should state that. If not, they need to clarify what they mean.

Thank you for the comments. We agree with you. To clarify what we mean, we modified the sentence (Line:80, 88).

2-2-7) Line 81 - Sentence would be improved by beginning 'Therefore, the question is whether...'

Line 120 - 'None of these backbone structures are similar...'

Line 154 - 'were found to be coherently packed to constitute...'

Thank you for pointing these out. We modified them (Line: 88, 132, 169).

2-2-8) Line 187 - 'which enables to locally change the conformations,...' needs to be reworded

We modified the discussion and deleted this sentence.

2-2-9) Line 676 - Figure legend should indicate that the CD spectra are recorded under high pressure

We modified it (Lines: 837-838, 1025).

2-2-10) Line 784 - Typo - change 'balky' to 'bulky'

Thank you for pointing this out. We modified it (Line: 963).

Decision Letter, first revision:

Message: 24th Oct 2022

Dear Dr. Koga,

Thank you again for submitting your manuscript "Design of complicated all- α protein structures". I apologize for the delay in responding, which resulted from the difficulty in obtaining suitable referee reports. This was due to the original reviewer #2 being no longer available to reevaluate the manuscript. We have recruited a new reviewer #2 to evaluate whether the original requests have been fulfilled in the revised version of the manuscript. The comments from the 2 reviewers who evaluated your paper are below. In light of those reports, we remain interested in your study and would like to see your response to the comments of the referees, in the form of a revised manuscript.

You will see that while most of the reviewer concerns from the first round of revision have been fulfilled, reviewer #2 notes that one protein is imperfectly folded and requests additional NMR experiments to characterize its dynamics and residue level stability, such as $\{^1\text{H}\}$ - ^{15}N NOE, R1, R2 and H/D hydrogen exchange. While we are reluctant to see manuscripts undergoing multiple rounds of revision, we agree with the reviewer that addressing these final comments, before we make a final decision, will improve the ms and increase its impact. Please be sure to address/respond to all concerns of the referees in full in a point-by-point response and highlight all changes in the revised manuscript text file. If you have comments that are intended for editors only, please include those in a separate cover letter.

We expect to see your revised manuscript within 6 weeks. If you cannot send it within this time, please contact us to discuss an extension; we would still consider your revision, provided that no similar work has been accepted for publication at NSMB or published elsewhere.

Reporting Summary:

When submitting the revised version of your manuscript, please pay close attention to our [href="https://www.nature.com/nature-portfolio/editorial-policies/image-integrity">Digital Image Integrity Guidelines. and to the following points below:](https://www.nature.com/nature-portfolio/editorial-policies/image-integrity)

Please note that all key data shown in the main figures as cropped gels or blots should be presented in uncropped form, with molecular weight markers. These data can be aggregated into a single supplementary figure item. While these data can be displayed in a relatively informal style, they must refer back to the relevant figures. These data should be submitted with the final revision, as source data, prior to acceptance, but you may want to start putting it together at this point.

Data availability: this journal strongly supports public availability of data. All data used in accepted papers should be available via a public data repository, or alternatively, as Supplementary Information. If data can only be shared on request, please explain why in your Data Availability Statement, and also in the correspondence with your editor. Please note that for some data types, deposition in a public repository is mandatory - more information on our data deposition policies and available repositories can be found below: <https://www.nature.com/nature-research/editorial-policies/reporting-standards#availability-of-data>

[redacted]

Sincerely,
Sara

Sara Osman, Ph.D.
Associate Editor
Nature Structural & Molecular Biology

Referee expertise:

Reviewers' Comments:

Reviewer #1:

Remarks to the Author:

I am satisfied that the Authors have addressed my comments/points and therefore recommend this for publication without further changes.

Reviewer #2:

Remarks to the Author:

The article "Design of complicated all-alpha protein structures" by Sakuma .. et al .. Koga is an impressed work in which all-alpha helical proteins with complicated folds are designed, produced and verified experimentally. There is one major concern, which regards the folding of H6_FoldU_Nomur" and several minor concerns.

Major concern.

It is an impressive achievement that almost all the proteins selected successfully folded into a structure which closely resembles the target design. However, for one protein, H6_FoldU_Nomur, the folding seems to be imperfect, since it shows: 1) a large slope in the pre-transition thermal denaturation baseline monitored by CD, 2) A relative low TM score (0.56) as reported in Sup. Fig. 9, 3) It has a low m-value and a high RMSD (3.1 Å) relative to the design and 4) in the HSQC spectrum (Sup. Fig. 12) in contrast to the spectra of the other designed proteins, this appears to be the only one that shows many (about 16) weak peaks (which are assigned to residues 8, 11, 18, 19, 20, 37, 39, 41, 43, 47, 45, 48, 77, 88 and 98). This suggests that not just the C-terminal helix but also other portions of the protein are not uniquely fixed in one structure but are in conformational exchange. To test this point and to better characterize this protein, it is recommended that the authors carry out additional NMR experiments to characterize its dynamics and residue level stability, such as $\{^1\text{H}\}$ - ^{15}N NOE, R1, R2 and H/D hydrogen exchange.

Minor comments:

1. Considering that globins as an example of a complex all helical protein fold are featured in the Abstract and Introduction of this MS, it would seem appropriate that the authors comment on the role of co-factors like heme in the folding of all helical proteins. The authors may also find it relevant to cite the classic paper on alpha helical proteins by Murzin and Finkelstein (1988) *J. Mol. Biol.* 204(3) 749-769.
2. The concept of "helix order" is interesting as are the histograms for helix order of natural proteins shown in Figure and Sup. Fig. 1. It would be interesting to know if ligand binding, such as to heme, affects the helix order.
3. page 5, line 105, Here it would be good to refer the reader to Fig. 2, top panel.
4. The equation for calculating the helix order should be included in the Methods section, not just the legend of Figure 1.
5. Page 5, lines 123-126: Could the authors elaborate more on how the five structures out of the myriad of possible designs were "manually selected" for production and characterization?
Also, on page 7, lines 142-143, how was manual selection performed?
6. Page 6, line 140 (last line) A sentence should not be started with a number. The

number should either be written out ("Ninety-one" instead "91") or the sentence should be re-written. Also "22" on line 334, page 14 should be written out.

7. Figure 6: What is the RMSD for H5_fold-0_Elsa (top right corner).

8. Please standardize the use of British (for example "grey") and American ("gray") English spellings.

9. Extended Data Figure 12. should be re-drawn because many labels are overlapped and it is impossible to read and determine whether the peaks in the C-terminal alpha helix of "H6-fold-U-Nomur" are present, weak or absent.

10. In the Methods section, please note that the His tags bind to Ni²⁺ (nickel II cation) and not nickel "metal". Therefore, the columns should be referred to as "Ni²⁺-affinity columns".

11. Page 20, line 483, Capitalize "B" in "ASLA biotec Ltd.).

12. Page 21, line 498. Considering substituting "trustful" by "trustworthy"

13. Page 21, line 510. Considering substituting "times" by "iterations"

14. Page 24, line 566. "G" in "Glycerol" should be in lower case.

Signed:
Douglas Vinson Laurents

Author Rebuttal, first revision:

To Reviewer #1:

I am satisfied that the Authors have addressed my comments/points and therefore recommend this for publication without further changes.

We sincerely appreciate your review.

To Reviewer #2:

We sincerely appreciate your comments and suggestions. We revised our manuscript according to them. Point-by-point responses are described in the following.

Major concern.

It is an impressive achievement that almost all the proteins selected successfully folded into a structure which closely resembles the target design. However, for one protein, H6_FoldU_Nomur, the folding seems to be imperfect, since it shows: 1) a large slope in the pre-transition thermal denaturation baseline monitored by CD, 2) A relative low TM score (0.56) as reported in Sup. Fig. 9, 3) It has a low m-value and a high RMSD (3.1 Å) relative to the design and 4) in the HSQC spectrum (Sup. Fig. 12) in contrast to the spectra of the other designed proteins, this appears to be the only one that shows many (about 16) weak peaks (which are assigned to residues 8, 11, 18, 19, 20, 37, 39, 41, 43, 47, 45, 48, 77, 88 and 98). This suggests that not just the C-terminal helix but also other portions of the protein are not uniquely fixed in one structure but are in conformational exchange. To test this point and to better characterize this protein, it is recommended that the authors carry out additional NMR experiments to characterize its dynamics and residue level stability, such as $\{^1\text{H}\}$ - ^{15}N NOE, R1, R2 and H/D hydrogen exchange.

Thank you for the suggestions. We carried out the ^{15}N - $\{^1\text{H}\}$ NOE, R1, R2, and CLEANEX-PM FHSQC experiments to examine the structural fluctuation for each residue of H6_fold-U_Nomur in detail. As you pointed out, not only the C-terminal helix, but also the interacting N-terminal helix were found to be fluctuated. We added a sentence to describe the results (Page 8, Lines 178-179) and the data to the Extended Data Fig. 20. We also add details of these experiments into the Methods section (Pages 24-25).

Minor comments:

1-1. Considering that globins as an example of a complex all helical protein fold are featured in the Abstract and Introduction of this MS, it would seem appropriate that the authors comment on the role of co-factors like heme in the folding of all helical proteins.

It is interesting to know the role of co-factors like heme in the folding of all helical proteins. The designed proteins in this paper, however, do not have the ability to bind co-factors such as heme. Therefore, it was difficult to comment on the impact of cofactors on the folding of helical proteins. We plan to design heme-binding proteins, which allows us to explore the role of heme in the folding.

1-2. The authors may also find it relevant to cite the classic paper on alpha helical proteins by Murzin and Finkelstein (1988) J. Mol. Biol. 204(3) 749-769.

Thank you. We cited this paper with one sentence to explain it (Page 4, Lines 79-80).

2. The concept of “helix order” is interesting as are the histograms for helix order of natural proteins shown in Figure and Sup. Fig. 1. It would be interesting to know if ligand binding, such as to heme, affects the helix order.

We agree with you that it is interesting to know if ligand binding affects the helix order. However, we realized that it is fundamentally difficult to study on this, because when there is a protein of which all experimentally solved structures do not bind a ligand, it is impossible to distinguish whether this protein has originally no ligand binding ability or it has ligand binding ability but the structures that bind the ligand have not been solved yet. In the future, we will investigate the relation between complicated structures and functions by creating functional proteins.

3. page 5, line 105, Here it would be good to refer the reader to Fig. 2, top panel.

Thank you for the suggestion. We referred it (Page 5, Line 107).

4. The equation for calculating the helix order should be included in the Methods section, not just the legend of Figure 1.

Thank you for the suggestion. We included the equation in the Methods section (Page 14, top).

5-1. Page 5, lines 123-126: Could the authors elaborate more on how the five structures out of the myriad of possible designs were “manually selected” for production and characterization?

Sorry for misleading you. We selected the five structures not manually but based on the criteria as described in the text that follows (Page 6, Lines 129-133). To make it clear, we deleted the “manually” and modified the sentence.

5-2. Also, on page 7, lines 142-143, how was manual selection performed?

We described how the selection was performed in the Methods section (Pages 16-17) and deleted “manual” to avoid confusion.

6. Page 6, line 140 (last line) A sentence should not be started with a number. The number should either be written out (“Ninety-one” instead “91”) or the sentence should be re-written. Also “22” on line 334, page 14 should be written out.

Following your suggestion, we wrote out the number in the both cases you pointed out (Page 7, Line 144; Page 14, Line 345).

7. Figure 6: What is the RMSD for H5_fold-0_Elsa (top right corner).

Following your suggestion, we showed the RMSD for H5_fold-0_Elsa in Figure 6.

8. Please standardize the use of British (for example “grey”) and American (“gray”) English spellings.

Thank you for pointing this out. We standardized our spelling to British English as long as we know.

9. Extended Data Figure 12. should be re-drawn because many labels are overlapped and it is impossible to read and determine whether the peaks in the C-terminal alpha helix of “H6-hold-U-Nomur” are present, weak or absent.

Following your suggestion, we re-drew Extended Data Figure 12.

10. In the Methods section, please note that the His tags bind to Ni²⁺ (nickel II cation) and not nickel “metal”. Therefore, the columns should be referred to as “Ni²⁺-affinity columns”.

Thank you for pointing this out. We modified it throughout the manuscript.

11. Page 20, line 483, Capitalize “B” in “ASLA biotec Ltd.).

We modified it (Page 21, Line 510).

12. Page 21, line 498. Considering substituting “trustful” by “trustworthy”

We modified it (Page 22, Line 525). Thank you.

13. Page 21, line 510. Considering substituting “times” by “iterations”

We modified it (Page 22, Line 537). Thank you.

14. Page 24, line 566. “G” in “Glycerol” should be in lower case.

We modified it (Page 27, Line 633). Thank you.

Decision Letter, second revision:

Message: Our ref: NSMB-A45429B

7th Apr 2023

Dear Dr. Koga,

Thank you for submitting your revised manuscript "Design of complicated all- α protein structures" (NSMB-A45429B). It has now been seen by the original referee who had had remaining concerns in the previous round of review and their comments are below. The reviewers find that the paper has improved in revision, and therefore we'll be happy in principle to publish it in Nature Structural & Molecular Biology, pending minor revisions to satisfy the referees' final requests and to comply with our editorial and formatting guidelines.

We are now performing detailed checks on your paper and will send you a checklist detailing our editorial and formatting requirements in a couple of weeks. Please do not upload the final materials and make any revisions until you receive this additional information from us.

To facilitate our work at this stage, it is important that we have a copy of the main text as a word file. If you could please send along a word version of this file as soon as possible, we would greatly appreciate it; please make sure to copy the NSMB account (cc'ed above).

Sincerely,
Sara

Sara Osman, Ph.D.
Associate Editor
Nature Structural & Molecular Biology

Reviewer #2 (Remarks to the Author):

The authors have addressed well and completely all the previous concerns I had regarding the first version of this MS.

Final Decision Letter:

Message Dear Dr Tatsumi-Koga,
:

Please find below a copy of the decision letter for your manuscript "Design of complicated all- α protein structures" [NSMB-A45429C], which has just been accepted for publication in Nature Structural & Molecular Biology.

The exact publication date will be communicated to the corresponding author. Please note that until publication, the content of your paper remains under embargo (to determine when the paper can be discussed with the media, please consult our embargo policy at http://www.nature.com/authors/editorial_policies/embargo.html).

If you wish to order reprints of your article or have any questions about reprints please send an email to author-reprints@nature.com.

Please contact the corresponding author directly with any queries you may have related to the content and publication of your paper.

As we prepare the manuscript for publication, we would like to confirm that your address details are correct. Could you please click on the link below to verify your profile and correct it as needed? Your prompt attention to this will help us to avoid delays in publication of your manuscript.

Please verify your address details promptly and correct them as needed by clicking here and following the link to "Login to My Account/Modify My NRG Profile":

[redacted]

Sincerely,
Sara Osman, Ph.D.
Associate Editor
Nature Structural & Molecular Biology

Subject: Decision on Nature Structural & Molecular Biology submission NSMB-A45429C

4th Oct 2023

Dear Dr. Koga,

We are now happy to accept your revised paper "Design of complicated all- α protein structures" for publication as an Article in Nature Structural & Molecular Biology.

Your paper will be published online soon after we receive proof corrections and will appear in print in the next available issue. You can find out your date of online publication by contacting the production team shortly after sending your proof corrections. Content is published online weekly on Mondays and Thursdays, and the embargo is set at 16:00 London time (GMT)/11:00 am US Eastern time (EST) on the day of publication. Now is the time to inform your Public Relations or Press Office about your paper, as they might be interested in promoting its publication. This will allow them time to prepare an accurate and satisfactory press release. Include your manuscript tracking number (NSMB-A45429C) and our journal name, which they will need when they contact our press office.

About one week before your paper is published online, we shall be distributing a press release to news organizations worldwide, which may very well include details of your work. We are happy for your institution or funding agency to prepare its own press release, but it must mention the embargo date and Nature Structural & Molecular Biology. If you or your Press Office have any enquiries in the meantime, please contact press@nature.com.

Please note that *Nature Structural & Molecular Biology* is a Transformative Journal (TJ). Authors may publish their research with us through the traditional subscription access route or make their paper immediately open access through payment of an article-processing charge (APC). Authors will not be required to make a final decision about access to their article until it has been accepted. <https://www.springernature.com/gp/open-research/transformative-journals> Find out more about Transformative Journals

In approximately 10 business days you will receive an email with a link to choose the

appropriate publishing options for your paper and our Author Services team will be in touch regarding any additional information that may be required.

Sincerely,
Sara

Sara Osman, Ph.D.
Associate Editor
Nature Structural & Molecular Biology
